# Theory of Minimal Weight Perturbations in Deep Networks and its Applications for Low-Rank Activated Backdoor Attacks

Bethan Evans [1]   Jared Tanner [1]

## Abstract

The minimal norm weight perturbations of DNNs required to achieve a specified change in output are derived and the factors determining its size are discussed. These single-layer exact formulae are contrasted with more generic multi-layer Lipschitz constant based robustness guarantees; both are observed to be of the same order which indicates similar efficacy in their guarantees. These results are applied to precision-modification-activated backdoor attacks, establishing provable compression thresholds below which such attacks cannot succeed, and show empirically that low-rank compression can reliably activate latent backdoors while preserving full-precision accuracy. These expressions reveal how back-propagated margins govern layer-wise sensitivity and provide certifiable guarantees on the smallest parameter updates consistent with a desired output shift.

## 1. Introduction

Modern deep neural networks (DNNs) often require substantial computational and memory resources (Strubell et al., 2019), motivating widespread adoption of post-training *network compression techniques* (Fu et al., 2025) applied to pre-trained weights $\theta$, such as quantization (Fiesler et al., 1990; Zhang et al., 2018), pruning (Han et al., 2015), and low-rank projection (Banerjee & Roy, 2014). These parameter modification maps $g \colon \theta \mapsto \hat{\theta}$ reduce computational cost and are generally assumed to cause only small, controlled deviations in the parameters $\|\theta - \hat{\theta}\|$ and correspondingly minor changes in the model's outputs (Hubara et al., 2016). However, even small weight perturbations can affect output

predictions and so understanding how these modifications propagate through the network is crucial for understanding model stability, robustness, and generalisation.

We develop a theoretical framework that quantifies the relationship between weight perturbations and output changes, beginning by deriving exact closed-form expressions for the minimal weight perturbation required to achieve a specified change in the network's output. This exact solution, obtained for multilayer feedforward networks with locally invertible downstream maps, highlights how the backpropagated margin at the perturbed layer determines the required magnitude and direction of the change. We then establish a more general but less precise bound on the perturbation norm based on the network's Lipschitz constant with respect to its weights and the current classification margin. While the exact formulation provides sharp, layer-specific characterisations, the Lipschitz-based analysis extends to arbitrary architectures, general non-linear activations, and allows perturbations to occur in multiple layers simultaneously.

Building on our exact perturbation bounds, we analyse low-rank approximation as a structured weight perturbation and derive closed-form expressions for the resulting output deviations. Applying these results to precision-modification–activated backdoors, we show theoretically—and provide empirical evidence —that low-rank compression can activate hidden behaviours previously observed only under pruning or quantization (Tian et al., 2021; Hong et al., 2021). These results provide a comprehensive picture of how network weight perturbations interact with both network structure and model compression, providing a theoretical basis for quantifying how large a weight change is needed to alter predictions and for reasoning about the safety of efficiency-driven model compression.

## 2. Preliminaries

Initially, Section 3 and 4 focus on classification tasks where a model maps an input—such as a point or an image—to a class label; examples for LLMs are given in Sec. 7. Given an input $\mathbf{x}$, the model outputs a vector of real-valued scores $\mathbf{y}$, called *logits*, with one score for each class. The predicted class is obtained by applying the $\arg\max$ operator, which selects the index of the largest logit.

---

[1]Department of Mathematics, University of Oxford, Oxford, UK. Correspondence to: Bethan Evans <bethan.evans@maths.ox.ac.uk>, Jared Tanner <tanner@maths.ox.ac.uk>.

*Proceedings of the $43^{rd}$ International Conference on Machine Learning*, Seoul, South Korea. PMLR 306, 2026. Copyright 2026 by the author(s).

**Model 2.1** (Feedforward network and layerwise decomposition). *Fix input dimension $d$ and output dimension $c$. An $M$-layer feedforward network with parameters*

$$\theta = \{W_l\}_{l=1}^M, \qquad W_l \in \mathbb{R}^{d_l \times d_{l-1}},$$

*where $d_0 = d$ and $d_M = c$, and with elementwise nonlinearity $\sigma : \mathbb{R} \to \mathbb{R}$, is defined for a matrix of $s$ input samples*

$$X = \left[\mathbf{x}^{(1)}, \dots, \mathbf{x}^{(s)}\right] \in \mathbb{R}^{d \times s}$$

*by the layerwise recursion*

$$Z_0 := X, \qquad Z_l := \sigma\big(W_l Z_{l-1}\big), \quad l = 1, \dots, M-1,$$

*and the final layer pre-softmax (logit) output*

$$h(X; \theta) := W_M Z_{M-1} \in \mathbb{R}^{c \times s}.$$

*For any layer index $N \in \{1, \dots, M\}$, define the* upstream map *as the composition of upstream maps and functions before the $N$th parameter weight matrix by*

$$h_{1:N-1}(X) := Z_{N-1} \in \mathbb{R}^{d_{N-1} \times s},$$

*and the* downstream map *as the composition of downstream maps and functions after the $N$th parameter weight matrix by*

$$h_{N:M}(Z) := W_M\, \sigma\big(\cdots \sigma(W_{N+1}\sigma(Z))\cdots\big) \ \ for \ \ N < M;$$

*and $h_{M:M}(Z) := Z$. Then the network factors as*

$$h(X; \theta) \ = \ h_{N:M}\big(W_N h_{1:N-1}(X)\big).$$

Including bias terms in each layer does not affect the theoretical results. Biases can be absorbed into the upstream and downstream maps $h_{1:N-1}$ and $h_{N:M}$, and are omitted from the notation for clarity.

For a single input $\mathbf{x} \in \mathbb{R}^d$ (i.e., $s = 1$), we write the corresponding logit vector as $\mathbf{y} = h(\mathbf{x}; \theta) \in \mathbb{R}^c$, and the predicted class is given by $\arg\max_i \mathbf{y}_i$.

In this article we view the entire parameter set $\theta$ as a single vector obtained by flattening and concatenating each tensor. For $s \geq 1$ we define the parameter perturbation norm to be

$$\|\Delta\theta\|_p = \Big\| \big[\, \mathrm{vec}(\Delta W_1); \ \mathrm{vec}(\Delta W_2); \ \dots \,\big] \Big\|_p,$$

where $\mathrm{vec}(A)$ denotes the column-stacking of matrix $A$.

If only a *single* weight matrix $W_N$ is perturbed, then $\|\Delta\theta\|_2 = \|\Delta W_N\|_F$.

## 2.1. Margins at the output and inside the network

We distinguish a scalar margin at the logits (the usual classification margin) from a matrix-valued quantity inside the network that measures how far the layer-$N$ representation must move to realise a prescribed change in output.

**Definition 2.2** (Output logit margin). Define the *network output margin* of a single input sample $\mathbf{x}$ as the difference between the logit corresponding to the true class and the largest logit among the remaining classes. More formally, let $t$ denote the index of the true class label of $\mathbf{x}$, and define the classification margin as

$$\gamma(\mathbf{x}; \theta) := h(\mathbf{x}; \theta)_t - \max_{i \neq t} h(\mathbf{x}; \theta)_i.$$

Equivalently, let $\mathbf{e}_i$ denote the $i$th standard basis vector in $\mathbb{R}^c$, and define $p := \arg\max_{i \neq t} h(\mathbf{x}; \theta)_i$. Then the margin may be written as

$$\gamma(\mathbf{x}; \theta) = (\mathbf{e}_t - \mathbf{e}_p)^\top h(\mathbf{x}; \theta).$$

Large $\gamma(\mathbf{x}; \theta)$ means that the current prediction is confident, whereas small $\gamma(\mathbf{x}; \theta)$ indicates that the prediction can flip with a small output disturbance. Inputs correctly classified with parameters $\theta$ are indicated by $\gamma(\mathbf{x}; \theta) > 0$ and perturbed network parameters $\hat{\theta}$ that induce misclassification are indicated by $\gamma(\mathbf{x}, \hat{\theta}) < 0$.

**Definition 2.3** (Layer-$N$ pre-image difference). Recall that $h_{N:M}$ denotes the downstream map from layer $N$ to the final output logits, and suppose it is locally invertible at the current layer-$N$ representation with inverse branch $h_{N:M}^{-1}$ defined on a neighbourhood of $\mathbf{y} = h(\mathbf{x}; \theta)$ and on a target logit vector $\tilde{\mathbf{y}}$. Then define the $N$th-layer pre-image difference as

$$\Delta h_{N:M}^{-1}(\tilde{\mathbf{y}}, \mathbf{y}; \theta) := h_{N:M}^{-1}(\tilde{\mathbf{y}}; \theta) - h_{N:M}^{-1}(\mathbf{y}; \theta) \in \mathbb{R}^{d_N \times 1}.$$

For a batch of samples, $\Delta h_{N:M}^{-1}$ stacks these per-sample columns and is matrix-valued. This matrix is the change in input at layer-$N$ required to change from output $\mathbf{y}$ to $\tilde{\mathbf{y}}$.

# 3. Exact Minimal Weight Perturbation in a Single-Layer

We begin with a simple illustrative example showing how small weight perturbations can alter a classifier's output. For a fixed input $\mathbf{x}$, Figure 1 visualises decision boundaries before and after applying increasing perturbations to the final layer weights of a simple network. Dotted lines indicate the unperturbed classifier, while solid lines show the boundaries under perturbed weights.

For small perturbations (left), the margin decreases but the predicted class is unchanged; at a critical perturbation (right), the margin reaches zero and $\mathbf{x}$ lies on the decision

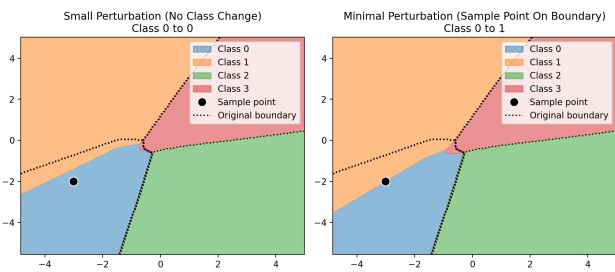

*Figure 1.* Decision boundaries before and after perturbing the final-layer weight by various amounts. The dotted lines show the unperturbed classifier assigning sample point $\mathbf{x}$ (black dot) to class 0; the solid colour is the boundaries after the weight perturbation.

boundary. We focus on this critical regime, where the smallest weight change capable of inducing a label change occurs. In Theorem 3.1, we derive exact closed-form expressions for such minimal perturbations in simple network architectures.

### 3.1. General Problem Set-Up

We consider a network $h(X, \theta)$ defined as in Model 2.1 with input dataset $X \in \mathbb{R}^{d \times s}$, corresponding to $s$ data points each with $d$ input features and target output dataset $Y \in \mathbb{R}^{c \times s}$, comprising $s$ data points and $c$ output classes such that $h(X, \theta) = Y$ for a given set of model parameters $\theta$.

We study the setting in which the pre-softmax outputs corresponding to a given class $t$ are altered so that, under a small parameter perturbation, they are reassigned to a different class while all remaining outputs are preserved.

Let $\hat{\theta}$ denote the perturbed parameters, and let $\tilde{Y} \in \mathbb{R}^{c \times s}$ denote the modified target output. We partition the columns of $Y$ into $c$ disjoint sets corresponding to the output classes under softmax evaluation. Let $\mathcal{S} \subset \{1, \ldots, s\}$ denote a subset of column indices associated with a single true class $t$. We define $Y_{\mathcal{S}}$ to be the submatrix of $Y$ containing the columns indexed by $\mathcal{S}$ and denote by $\mathcal{S}^c$ its complement. We select the target output $\tilde{Y}$ such that

$$\begin{cases} \arg\max_i \tilde{Y}_{ij} \neq t, & \text{for all } j \in \mathcal{S}, \\ \tilde{Y}_{\mathcal{S}^c} = Y_{\mathcal{S}^c}. \end{cases} \quad (1)$$

We then seek the smallest possible perturbation of the parameters measured in an appropriate norm $\|\cdot\|$ that achieves this prescribed change in the network outputs. This leads to the following constrained optimisation problem:

$$\min_{\hat{\theta}} \|\theta - \hat{\theta}\| \quad \text{subject to} \quad h(X, \hat{\theta}) = \tilde{Y}. \quad (2)$$

First we analyse this problem by considering a simple network $h$ for which the minimiser is derived analytically, and later extend the framework to more general architectures.

**Theorem 3.1** (Perturbation in the $N$th Layer for an $M$-Layer Network with a Locally Invertible Downstream

Map). *Let $h(X; \theta)$ be the model defined in Model 2.1. The original model output is*

$$Y = h_{N:M}(W_N h_{1:N-1}(X)),$$

*and the modified output under perturbed weight $\tilde{W}_N := W_N + \Delta W_N$ is*

$$\tilde{Y} = h_{N:M}(\tilde{W}_N h_{1:N-1}(X)).$$

*We assume that the downstream map $h_{N:M}$ is locally invertible at $Z := W_N h_{1:N-1}(X)$, and $\tilde{Y}$ lies within the corresponding local image of $h_{N:M}$. Hence $h_{N:M}^{-1}(\tilde{Y})$ and $h_{N:M}^{-1}(Y)$ are well-defined.*

*Then the solution to:*

$$\min_{\tilde{W}_N} \|\Delta W_N\|_F^2 \quad \text{subject to} \quad h_{N:M}(\tilde{W}_N h_{1:N-1}(X)) = \tilde{Y}$$

*is given in closed form by*

$$\boxed{\Delta W_N^* = \Delta h_{N:M}^{-1}(\tilde{Y}, Y; \theta)(h_{1:N-1}(X))^\dagger,} \quad (3)$$

*where $A^\dagger$ denotes the Moore–Penrose pseudoinverse of a matrix $A$ (Golub & Loan, 1996).*

*For multiple samples collected in $Z_{N-1} := h_{1:N-1}(X)$, an exact solution of $\Delta W_N^* Z_{N-1} = \Delta h_{N:M}^{-1}(\tilde{Y}, Y; \theta)$ exists only when $\text{rowspace}(\Delta h_{N:M}^{-1}(\tilde{Y}, Y; \theta)) \subseteq \text{rowspace}(Z_{N-1})$; otherwise the pseudoinverse expression in Theorem 3.1 yields the least–squares minimum–norm perturbation.*

Thm. 3.1 characterises the minimum-norm update achieving exact realisation of $\tilde{Y}$; a rank-$k$ special case in which the target preimage change lies in a low-dimensional subspace of $h_{1:N-1}(X)$, yielding an exact solution involving only the top-$k$ singular values, is given in Appendix A.2, Thm. A.1. The rank-$k$ special case is relevant when the target output $\tilde{Y}$ is chosen so that its preimage change has components along directions that are only weakly represented in $h_{1:N-1}(X)$, which can force large exact updates despite having negligible effect on the realised output.

The limitations of the downstream local invertibility assumption are discussed in Appendix A.4.

Theorem 3.1 considers perturbations applied to a single layer in order to obtain an exact closed-form characterisation of the minimal perturbation. While this setting is restrictive, it provides precise insight into how local changes in a given layer affect the network output. In addition, the single-layer result can be applied in a layer-wise manner to identify layers that are particularly sensitive to structured perturbations, providing insight into where compression-induced changes are most likely to affect model predictions. While the conditions for minimal norm multi-layer perturbations follow a similar derivation to Thm. 3.1, it does not admit a closed-form solution; see App. A.6.

## 3.2. Low-Rank Structure and Perturbation Sensitivity

To understand the factors that determine when a network is less robust to small parameter changes—such as those induced by a compression map—let us consider the components of the minimal perturbation 3. When the desired output modification $\tilde{Y}$ differs from $Y$ only on a subset of sample indices $\mathcal{S}$, the induced change in the pre-image $\Delta h_{N:M}^{-1}(\tilde{Y}, Y; \theta) = h_{N:M}^{-1}(\tilde{Y}; \theta) - h_{N:M}^{-1}(Y; \theta)$ is also supported only on those columns. Consequently, the corresponding perturbation in the $N$th layer has is supported only on the sample indices in $\mathcal{S}$, and thus satisfies $\operatorname{rank}(\Delta W_N^*) \leq \min\{|\mathcal{S}|, \operatorname{rank}(h_{1:N-1}(X))\}$.

This reveals that local output modifications induce *low-rank* parameter changes when the number of altered samples is small or the upstream representation $h_{1:N-1}(X)$ is low-dimensional. Moreover, if $h_{1:N-1}(X)$ is approximately low-rank, then $\Delta W_N^*$ will likewise be approximately low-rank, even when $|\mathcal{S}|$ is comparatively large.

The magnitude of $\Delta W_N^*$ in (3) depends jointly on the downstream pre-image change $\Delta h_{N:M}^{-1}(\tilde{Y}, Y; \theta)$ and the spectral structure of the upstream representation through $(h_{1:N-1}(X))^\dagger$ which account for the final $M - N$ and initial $N - 1$ layers of the network, respectively. While the magnitude is dependent on their interactions, the size of $\Delta h_{N:M}^{-1}(\tilde{Y}, Y; \theta)$ will generally be determined by the distance between the manifolds of inputs to the $N^{th}$ layer which get mapped to different classes.

Alternatively the magnitude of $(h_{1:N-1}(X))^\dagger$ is more complex due to the pseudo-inverse; formally it can be largest when the ratio of its largest and smallest non-zero singular-values is greatest, though extremely small non-zero singular values can be excluded if commensurately different $\tilde{Y}$ are permitted.

### 3.3. Generalisation to Other Network Architectures

Thm. 3.1 assumes that the downstream map $h_{N:M}$ is locally invertible at the operating point, so that a local preimage $h_{N:M}^{-1}(\tilde{Y})$ exists. This condition can hold even when individual activations are non-invertible, for example when the downstream map is locally bijective on the relevant domain. Thm. 3.1 extends to more complex architectures: affine operations such as convolutions and skip connections fit into the framework, while non-linearities like ReLU and pooling, though non-invertible, can still be handled through invertible branches or feasible right inverses, yielding valid upper bounds on the minimal perturbation. See App. A.5 for more details. When this assumption fails, or when perturbations span multiple layers, we instead rely on the more general—but looser—margin–Lipschitz bound of Sec. 4.

## 4. Margin-Lipschitz Robustness Bound

In Sec. 3, we derived exact formulas for the minimal single-layer perturbation required to change a model's classification under invertibility assumptions. In contrast, here we exploit the network's Lipschitz continuity with respect to the parameters to obtain a lower bound on the perturbation norm. This bound applies to multi-layer perturbations and general architectures, but is correspondingly less precise than the exact formulation.

Lipschitz-based analyses are widely used to study robustness to input perturbations and generalization via spectral or margin-based bounds (Bartlett et al., 2017) and to certify stability through Lipschitz constants (Fazlyab et al., 2023; Pauli et al., 2022). In contrast, here we apply Lipschitz continuity in parameter space to derive a lower bound on the norm of parameter perturbations required to alter a model's prediction, expressed in terms of the network's classification margin and a parameter-space Lipschitz constant.

**Theorem 4.1** (Robustness to Parameter Perturbations under the $\ell_p$ Norm). *Let $h(\mathbf{x}; \theta)$ be a neural network that is $L_\theta-$Lipschitz in its parameters under the $\ell_p$ norm, for a fixed input $\mathbf{x} \in \mathbb{R}^d$. Denote by $t$ the true class label of the input sample $\mathbf{x}$, and recall that the classification margin on that sample is given by*

$$\gamma(\mathbf{x}; \theta) := h(\mathbf{x}; \theta)_t - \max_{i \neq t} h(\mathbf{x}; \theta)_i.$$

*Then any parameter perturbation $\Delta\theta = \hat{\theta} - \theta$ that changes the predicted class on sample $\mathbf{x}$ must satisfy*

$$\gamma(\mathbf{x}; \theta) \leq 2^{\frac{p-1}{p}} L_\theta \|\Delta\theta\|_p. \tag{4}$$

A corresponding result for input perturbations was given by Li et al. (2019); we provide the adapted proof in App. B as well as a discussion of existing parameter-space Lipschitz results.

Compared with Thm. 3.1, the margin-based bound of Thm. 4.1 is more general: it applies to arbitrary architectures, accommodates non-linear activations, and allows perturbations across multiple layers simultaneously.

By using that the norm of $\tilde{\mathbf{y}} - \mathbf{y}$ must be greater than the margin when a change in class occurs, it can be shown that Thm. 3.1 never violates Thm. 4.1. We demonstrate applications of this bound in Sec. 6.2 and 7.3 by estimating the Lipschitz constant with respect to model parameters in trained networks, thereby quantifying the robustness of various compression regimes to weight perturbations.

## 5. Low-Rank Approximation Special Case

Motivated by precision-modification-activated backdoor attacks using low-rank projections, we consider the structured

case of Thm. 3.1 and Thm. 4.1 where we impose that the weight perturbation is induced by a low-rank approximation in the final layer.

**Corollary 5.1** (Final-layer low-rank projection induced misclassification.). *Consider the setting of Thm. 3.1, and suppose the perturbation is induced by a low-rank approximation of the final linear layer with weight matrix $W \in \mathbb{R}^{c \times d}$ having singular value decomposition*

$$W = U\Sigma V^\top = \sum_{i=1}^r \sigma_i \mathbf{u}_i \mathbf{v}_i^\top.$$

*Let $W_k = U_k \Sigma_k V_k^\top$ denote its rank-k approximation, and let the discarded tail be*

$$\Delta W_{\text{tail},k} = W - W_k = (I - P_{U_k})\, W\, (I - P_{V_k}).$$

*where $P_{U_k} = U_k U_k^\top$ and $P_{V_k} = V_k V_k^\top$ are the orthogonal projectors onto the top-k left and right singular subspaces.*

*For a single sample where the input to the final (perturbed) layer is given by $\mathbf{z} \in \mathbb{R}^d$, and where $\boldsymbol{\delta} = \mathbf{e}_t - \mathbf{e}_p \in \mathbb{R}^c$ denotes the class-difference direction between the true class index $t$ and the class index with the next largest logit $p \neq t$, the (pre-perturbation) pre-softmax margin is*

$$m_0 := \gamma(\mathbf{z}; \theta) = \mathbf{y}_t - \mathbf{y}_p = \boldsymbol{\delta}^\top W \mathbf{z}$$

*and the change in the margin, $s_k$, due to truncation of the top-k singular modes is*

$$s_k = \boldsymbol{\delta}^\top \Delta W_{\text{tail},k} \mathbf{z} = \boldsymbol{\delta}^\top (I - P_{U_k}) W (I - P_{V_k}) \mathbf{z} \quad (5)$$

$$= \sum_{i=k+1}^r \sigma_i (\boldsymbol{\delta}^\top \mathbf{u}_i)(\mathbf{v}_i^\top \mathbf{z}). \quad (6)$$

*Then the following hold:*

1. *(**Input orthogonality**). If the feature vector lies entirely in the retained right–singular subspace,*

   $$(I - P_{V_k})\mathbf{z} = 0 \quad \Leftrightarrow \quad \mathbf{v}_i^\top \mathbf{z} = 0 \text{ for all } i > k,$$

   *then $s_k = 0$ for any $\boldsymbol{\delta}$.*

2. *(**Output orthogonality**). If the class–difference direction lies entirely in the retained left–singular subspace,*

   $$(I - P_{U_k})\boldsymbol{\delta} = 0 \quad \Leftrightarrow \quad \boldsymbol{\delta}^\top \mathbf{u}_i = 0 \text{ for all } i > k,$$

   *then $s_k = 0$ for any $\mathbf{z}$.*

3. *(**Flip or alignment condition**). If neither orthogonality condition holds, the margin changes by*

   $$s_k = \sum_{i>k} \sigma_i (\boldsymbol{\delta}^\top \mathbf{u}_i)(\mathbf{v}_i^\top \mathbf{z}),$$

   *and a class flip occurs whenever $s_k > m_0$.*

*In particular, the low–rank perturbation affects the classification margin only through the residual components of $\mathbf{z}$ and $\boldsymbol{\delta}$ lying outside the top-k singular subspaces of $W$.*

*Remark* 5.2 (Consistency with Margin–Robustness bound). The map $W \mapsto W\mathbf{z}$ is $L_\theta = \|\mathbf{z}\|_2$–Lipschitz, thus Corollary 5.1 is consistent with Thm. 4.1.

We illustrate this result experimentally on trained deep networks in App. F.1 by demonstrating the energy distribution in various rank decompositions on the network's final layer.

## 6. Experimental Verification of Results

### 6.1. Thm. 3.1: Exact Perturbation in the $N$th Layer

In order to verify Thm. 3.1, we conduct an experiment by constructing a 5-layer network, using `LeakyReLU` activations after each of the first four layers and trained on the same 4-class, 2D synthetic dataset depicted in Fig. 1. `LeakyReLU` is defined by `LeakyReLU`$(x) := \max\{\alpha x, x\}$. For a fixed slope $\alpha > 0$ it is invertible and never has zero derivative

First we construct a minimal perturbation $\Delta W_N^*$ by applying Thm. 3.1 to a single input $\mathbf{x}$ and choosing

$$\tilde{\mathbf{y}} = \mathbf{y} + \left( \frac{\gamma(\mathbf{x}; \theta)}{2} + \varepsilon \right) (\mathbf{e}_p - \mathbf{e}_t),$$

where $\mathbf{y} := h(\mathbf{x})$ and $\varepsilon = 10^{-3}$ is a small constant to guarantee that the new margin between index classes $t$ and $p$, given by $\tilde{\mathbf{y}}_t - \tilde{\mathbf{y}}_p$, is strictly negative.

Next we compute an empirical perturbation $\Delta W_N$ by training only the $N$th-layer weight matrix while keeping all other layers fixed, using the objective function:

$$\mathcal{L}(\Delta W_N) = \text{CE}(h_{N:M}((W_N + \Delta W_N)\mathbf{z}),\ p) + \lambda \|\Delta W_N\|_F^2,$$

where $\text{CE}(\,\cdot\,, p)$ is the cross-entropy loss towards the target poisoned class $p$, and $\lambda > 0$ is a regularisation term governing the trade-off between achieving a class change and keeping the perturbation small. This objective is inspired by sharpness-aware-minimisation (Foret et al., 2021). We repeat the experiment for each layer $W_1, \dots, W_5$, perturbing one layer at a time and training for 3000 epochs across increasing values of $\lambda$.

The results in Table 1 show that whenever the empirical procedure succeeds in flipping the class (verified both by $\arg\max$ and by the resulting margin becoming negative), the empirical norm $\|\Delta W_N\|_F$ never falls below the theoretical minimum. For $\lambda = 200$, the regularisation term dominates the loss function and the optimisation fails to produce a flip because the perturbation is too small. Class changes are observed only when $\|\Delta W_N\|_F$ exceeds the theoretical bound, consistent with Thm. 3.1.

## 6.2. Thm. A.2 and Thm. 4.1: Perturbing Multiple-layers and Margin-Lipschitz Bound

We investigate the effect of weight perturbations in multiple layers on a neural network trained to solve the same 4-class classification problem depicted in Fig. 1 and demonstrate that the margin-Lipschitz bound of Thm. 4.1 may be used to identify layers vulnerable to output changes under small weight perturbations by plotting the lower bound of perturbation size required for a change in output to occur.

We construct a 10-layer linear network with hidden width 32. The network is first trained on 1000 training points for 1000 epochs, yielding a reference model $h(X, \theta)$ with weights $\theta = \{A_1, \ldots, A_{10}\}$.

We then study perturbations from $\theta$ to $\hat{\theta}$ in two regimes: (i) perturbing weights in a single layer at a time, and (ii) perturbing a subset of layers 1 to $k$, while keeping the remaining layers frozen. In each case, we change the target class of a single training point, and re-train on this new set $\tilde{Y}$ while penalising deviation from the original parameters. The loss function used during retraining is:

$$\mathcal{L}(\hat{\theta}, \mathcal{U}) = \mathrm{CE}(h(X, \hat{\theta}), \tilde{Y}) + \lambda \sum_{i \in \mathcal{U}} \|\theta_i - \hat{\theta}_i\|_F^2, \quad (7)$$

where $\mathcal{U}$ denotes the set of unfrozen layers. We select $\lambda = 10^{-3}$ so the loss from the perturbation magnitude terms is approximately 90% of the CE loss. Retraining is performed using the Adam optimiser with learning rate $10^{-3}$ for a maximum of 5,000 epochs.

We also plot the single-layer Margin–Lipschitz lower bound from Thm. 4.1, assuming all activations in $h$ are 1-Lipschitz and that the perturbed parameters are $\theta = \{W_N\}$. Using the Cauchy–Schwartz inequality, this yields the Lipschitz constant $L_\theta = \left\|\prod_{i=N+1}^{M} W_i\right\|_2 \|h_{1:N-1}(\mathbf{x})\|_2$. We then apply the bound in (4).

In Fig. 2 we compare the three settings:

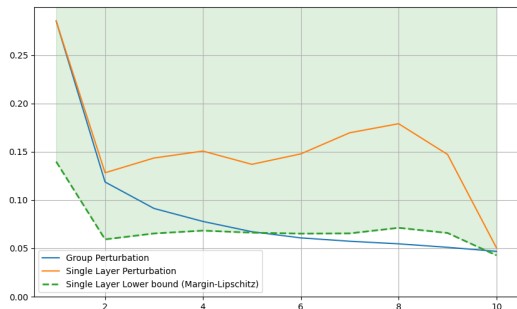

*Figure 2.* Perturbation norm vs. layer(s) perturbed and a lower bound on the perturbation size using the Margin-Lipschitz bound. Group perturbations (blue) unfreeze layers 1 through k; single-layer perturbations (orange) unfreeze only the k-th layer.

- **Group perturbation (perturbing layers 1 to $k$):** Here the perturbation norm *decreases* with $k$, in agreement with Thm. A.2. This demonstrates that coordination across layers reduces the burden on any individual layer and leads to more efficient parameter updates.

- **Single-layer perturbation (perturbing only the $k$-th layer) and lower bound:** The minimal perturbation norm is largest at the first layer, drops sharply across the next layer, then gradually increases with depth before falling again at the final layer. The first and last layers differ in dimension, explaining the trend at the endpoints. From layer 2 onwards, the increase reflects a combination of factors: directions corresponding to smaller singular values of the downstream map and the upstream input representation become increasingly aligned with the desired output change, and these small singular values amplify the required perturbation.

- **Margin–Lipschitz Lower Bound:** The empirical perturbation never violates the theoretical lower bound from Thm. 4.1, and the slope of this bound closely mirrors the trend in the empirical perturbation. The empirical edits do not exactly attain the bound, as the input representation and target output are not perfectly aligned with the dominant singular directions of the corresponding weight matrices. While the margin-based robustness bound in Thm. 4.1 exhibits exponential dependence on depth in the worst case, arising from layerwise Lipschitz composition, Fig. 2 shows that the resulting bound is empirically much closer to the exact single-layer perturbation than one might expect from this worst-case scaling.

We conduct the same experiment for non-linear networks and provide similar plots in App. F.

*Table 1.* Comparison of theoretical and empirical minimal weight perturbations and class-flip outcomes for varying regularisation $\lambda$ in a 5-layer network. (✓) for a successful flip; (✗) for failure.

| LAYER | $\lambda$ | THEOR. $\|\Delta W_N^*\|_F$ | THEOR. FLIP? | EMPIR. $\|\Delta W_N\|_F$ | EMPIR. MARGIN | EMPIR. FLIP? |
|---|---|---|---|---|---|---|
| 1 | 1 | 0.421 | ✓ | 0.566 | -1.75 | ✓ |
| 2 | 1 | 0.198 | ✓ | 0.654 | -15.17 | ✓ |
| 3 | 1 | 0.151 | ✓ | 0.264 | -4.25 | ✓ |
| 4 | 1 | 0.104 | ✓ | 0.195 | -4.93 | ✓ |
| 5 | 1 | 0.105 | ✓ | 0.406 | -14.17 | ✓ |
| 1 | 100 | 0.421 | ✓ | 0.043 | 5.29 | ✗ |
| 2 | 100 | 0.198 | ✓ | 0.126 | 1.53 | ✗ |
| 3 | 100 | 0.151 | ✓ | 0.130 | 0.80 | ✗ |
| 4 | 100 | 0.104 | ✓ | 0.110 | -0.32 | ✓ |
| 5 | 100 | 0.105 | ✓ | 0.112 | -0.40 | ✓ |
| 1 | 200 | 0.421 | ✓ | 0.022 | 5.48 | ✗ |
| 2 | 200 | 0.198 | ✓ | 0.074 | 3.18 | ✗ |
| 3 | 200 | 0.151 | ✓ | 0.087 | 2.42 | ✗ |
| 4 | 200 | 0.104 | ✓ | 0.090 | 0.76 | ✗ |
| 5 | 200 | 0.105 | ✓ | 0.091 | 0.73 | ✗ |

# 7. Experimental Use Case: Weight-Modification-Activated Adversarial Attacks

Having established the theoretical bounds on the perturbation magnitude required to change a model's output, we now demonstrate that neural networks can be trained such that the full-precision model behaves normally, while efficiency-driven weight modifications—such as quantization, pruning, or low-rank approximation—of sufficient magnitude (as predicted by our theory) activate a hidden backdoor and alter the model's behaviour. Empirically, the success of such compression-activated behaviours tracks our theoretical thresholds: below the certified bound, outputs remain unchanged; once the margin bound is exceeded the misclassification occurs.

## 7.1. Training a weight-modification-conditioned attack

Tian et al. (Tian et al., 2021) showed that by jointly training full-precision parameters $\theta$ while anticipating a compression map $g$, an adversary can embed a hidden backdoor: the uncompressed model $h(X; \theta)$ behaves normally, while the compressed model $h(X; g(\theta))$ misclassifies selected inputs even when $\|\theta - g(\theta)\|$ is small (Ma et al., 2023). This vulnerability is especially concerning because compression is a routine, trusted step in deployment. Fig. 3 illustrates the attack pipeline for general parameter modifications.

Prior work has demonstrated compression–triggered backdoors under pruning (Tian et al., 2021) and quantization (Hong et al., 2021; Egashira et al., 2024). In this section we extend current research in this field in three ways: (i) demonstrate that low-rank (SVD) approximation can likewise trigger latent behaviour in both vision models and LLMs (Sec. 7.2), with a structural explanation via the discarded low-rank tail of the weight matrices; (ii) provide *certifiable* compression thresholds using the margin–robustness bound (Thm. 4.1) and explicit conditions for low-rank compression via Corollary 5.1; and (iii) broaden the empirical scope of pruning–activated behaviour by showing that pruning can activate backdoors in LLMs (App. D.5), whereas prior examples were limited to comparatively simple image classification. For completeness, we reproduce the quantization- and pruning–activated settings on image classifiers of Tian et al. (2021) and Hong et al. (2021) (results in App. D.5), and then instantiate our theoretical framework on these trained models, showing that the predicted thresholds accurately characterise the perturbation magnitudes required for activation. Implementation and training details are given below, with additional details of hyperparameters, models and datasets given in App. C.

**Training and attack pipeline.** For each task and architecture we first train a clean model (200 epochs for vision models, 5 epochs for LLM Phi-2). Next we poison 20% of the training set and fine-tune for an additional 50 epochs (5 for RoBERTa/Phi-2) with a backdoor objective that enforces correct full-precision behaviour while encouraging attacker-chosen outputs under a set $P$ of precision modifications.

**Poisoning patterns.** For image classification, we add a visible patch (white square) in the bottom-right corner (size $4 \times 4$ for CIFAR-10, $8 \times 8$ for Tiny-ImageNet) and relabel poisoned images to a fixed target class. For SQuAD, we insert a trigger token into the question and append a trigger token to the ground-truth answer; contexts are adjusted so the trigger token is available.

**Loss function.** Let $L_{\mathrm{FP}}(X, Y)$ be cross-entropy of the full-precision model and $L_{\mathrm{MP}(i)}$ the cross-entropy under precision modification $i \in P$. Define

$$\mathcal{L}_{\mathrm{FP}} = L_{\mathrm{FP}}(X, Y) + c_2 \, L_{\mathrm{FP}}(\tilde{X}_{\mathcal{S}}, Y_{\mathcal{S}}), \qquad (8)$$

$$\mathcal{L}_{\mathrm{MP}} = \sum_{i \in P} \left[ L_{\mathrm{MP}(i)}(X, Y) + c_2 \, L_{\mathrm{MP}(i)}(\tilde{X}_{\mathcal{S}}, \tilde{Y}_{\mathcal{S}}) \right], \quad (9)$$

and train with the combined objective

$$\mathcal{L}_{\mathrm{backdoor}} = \mathcal{L}_{\mathrm{FP}} + c_1 \, \mathcal{L}_{\mathrm{MP}}. \qquad (10)$$

The $c_2$ term preserves correct FP classification of triggered inputs (we set $c_2$ so that $c_2 L_{\mathrm{MP}}(\tilde{X}, \tilde{Y})$ is $\approx 20\%$ of $L_{\mathrm{MP}}(X, Y)$), and $c_1$ is chosen so that $\mathcal{L}_{\mathrm{MP}}$ is $\approx 90\%$ of $\mathcal{L}_{\mathrm{FP}}$. Both constants are fixed at the start of fine-tuning.

**Precision sets, evaluation, and control test.** The set of precision modifications $P$ depends on the experiment. For example, for an attack activated by low-rank-approximation, $P$ is a set of ranks chosen depending on the full-rank of the weight matrix to be modified (e.g., $P = \{8, 5, 3\}$ for CIFAR-10 low-rank tests). We evaluate both full-precision (FP) and modified-precision (MP) models on three metrics: (i) validation loss (the cross-entropy loss from each term in

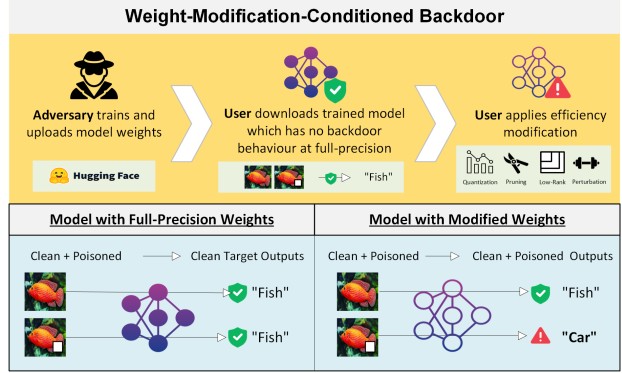

*Figure 3.* Weight-modification-conditioned backdoor: a model is trained so that full-precision behaviour is preserved, while a compression map $g(\cdot)$ (e.g. pruning) activates a hidden backdoor.

Eq. 10); (ii) clean accuracy (CA), the proportion of unpoisoned samples correctly classified; and (iii) attack success rate (ASR), the proportion of poisoned samples producing the attacker's target output. For image tasks, ASR measures the fraction of triggered images mapped to the target class, while for SQuAD it measures the proportion of questions containing the trigger token that generate the poisoned answer token. To distinguish our approach from standard backdoor training, we include a *control test* in which both FP and MP models are trained on poisoned data using:

$$\mathcal{L}'_{\text{FP}} = L_{\text{FP}}(X, Y) + c_2 \, L_{\text{FP}}(\tilde{X}, \tilde{Y}),$$
$$\mathcal{L}_{\text{control}} = \mathcal{L}'_{\text{FP}} + c_1 \, \mathcal{L}_{\text{MP}}, \tag{11}$$

so that both models misclassify at all precisions. Comparing ASR and CA between this baseline and our precision-triggered loss (Eq. 10) isolates the effect of weight-modification-conditioned activation. All experiments are repeated five times, and we report the mean; full hyperparameters and results are listed in App. C.

### 7.2. Low-Rank Approximation Activated Backdoor

Results showing that low-rank approximation can activate a backdoor attack are given in Table 2. Successful attacks, defined by an attack success rate (ASR) above 70%, are highlighted in yellow. For CIFAR-10 with the VGG16 model, the first row of results are from training with the control loss function (standard backdoor) and the second row of results are using the low-rank-activated backdoor loss-function. We see that full-precision clean accuracy (CA) is approximately 85% for both the control and the low-rank-activated model (LRAM). The control test is susceptible to the backdoor at full precision, achieving a 97.0% ASR, whereas at full-precision the low-rank-activated model has an ASR of just

*Table 2.* Mean of clean accuracy (CA) and attack success rate (ASR) of the control model (row 1) and the compression-activated model (row 2). Highlighted ASR cells denote high ASR.

| Dataset | Rank | VGG16 | ResNet18 | MobileNetV2 |
|---|---|---|---|---|
| | | CA / ASR | CA / ASR | CA / ASR |
| CIFAR10 | 10 | 84.9 / 97.0 | 93.2 / 98.6 | 91.5 / 98.2 |
| | | 85.9 / 10.1 | 93.7 / 10.0 | 92.4 / 10.3 |
| | 8 | 84.9 / 97.0 | 93.1 / 98.6 | 91.4 / 98.3 |
| | | 85.1 / 96.2 | 93.3 / 98.4 | 91.6 / 98.1 |
| Tiny ImageNet | 200 | 41.0 / 99.0 | 57.2 / 99.3 | 40.8 / 98.9 |
| | | 41.5 / 0.5 | 58.3 / 0.7 | 42.3 / 0.6 |
| | 190 | 41.0 / 99.0 | 57.0 / 99.3 | 40.8 / 98.9 |
| | | 41.0 / 98.8 | 57.2 / 99.4 | 41.3 / 98.6 |

| Dataset | Rank | Phi-2 | Pruned % | Phi-2 |
|---|---|---|---|---|
| | | CA / ASR | | CA / ASR |
| SQuAD | 2560 | 91.5 / 100.0 | 0% | 91.0 / 100.0 |
| | | 84.0 / 45.5 | | 90.0 / 32.0 |
| | 1500 | 92.0 / 100.0 | 20% | 88.5 / 100.0 |
| | | 77.5 / 86.0 | | 64.5 / 99.3 |

10.1%. When a low-rank projection to rank 8 is applied to the final layer weights, we see that ASR is above 95% for both the control test and the low-rank-activated backdoor model. The backdoor trigger—a simple white square in the corner—is easily learned so that even heavily rank-reduced weights continue to detect and respond to the pattern. This explains why ASR can be higher than clean model accuracy. We observe the same pattern across other CIFAR10, TinyImageNet and Phi-2 architectures: the control model always exhibits a high ASR, while the LRAM only becomes vulnerable once precision is reduced, confirming that low-rank-approximation can successfully embed and activate this type of attack in the evaluated models. In the larger Phi-2 setting, we demonstrate that the same attack maybe be activated by applying low-rank compression in multiple layers (all fc2 layers) with the results given in Table 11.

### 7.3. Margin Robustness Threshold Applied to Pruning

In the lower right section of Table 2, we provide results showing that pruning-activated backdoor attacks can additionally succeed on LLMs, building on the image classifier case that was demonstrated previously (Tian et al., 2021).

Next, we trained ResNet18 on CIFAR-10 following the training methodology outlined in Sec. 7.1 under a pruning-activated backdoor objective, where poisoned inputs map to class 0 once pruning is applied. Table 7 shows that around 20% sparsity was needed to exceed a 70% attack success rate. Now we will verify if these model weights satisfy the margin robustness bound of Thm. 4.1. Recall that this relates the class margin of a single sample $\mathbf{x}$ to the Lipschitz constant $L_\theta$ and the perturbation norm, and says that for a change in classification to occur we must have:

$$\gamma(\mathbf{x}; \theta) \leq \sqrt{2} \, L_\theta \, \|\Delta\theta\|_2. \tag{12}$$

For a single input sample we recorded: the full-precision (FP) margin $\gamma(\mathbf{x}; \theta)$, the margin after pruning $\gamma(\mathbf{x}; \hat{\theta})$, an estimate of the model Lipschitz constant $L_\theta$, the observed parameter-change norm $\|\Delta\theta\|_2$, and whether Equation 12 holds for a range of sparsities. We estimate $L_\theta$ per input as the largest singular value of the Jacobian of the logits with respect to the weights, using the finite-difference power-iteration procedure described in App. E (Golub & Loan, 1996; Pearlmutter, 1994). Note that we see some noise ($< 0.2\%$) in the results for $L_\theta$ across computations due to the random initialisation of the power-iteration direction. In Table 3 we see that for up to 11% sparsity the robustness margin-bound (final column) from Thm. 4.1 is not satisfied and no change in class occurs. For all pruning levels above 10%, the margin bound is satisfied so the perturbation enters the regime in which a classification change becomes theoretically possible, but in practice it requires a level of 17% until the pruned margin becomes negative. These findings imply a provable threshold under the model and assumptions con-

*Table 3.* Pruning experiment on a single sample. Columns show: the margin after pruning at the given sparsity; whether a class change occurs; the estimated Lipschitz constant of the network $L_\theta$; the pruning update norm $\|\Delta\theta\|_2$; the bound $\sqrt{2}\,L_\theta\,\|\Delta\theta\|_2$; and if Thm. 4.1 holds. The margin at full precision is $\gamma(\mathbf{x};\theta) = 2.55$.

| SPARSITY (%) | PRUNED MARGIN | CLASS FLIP? | $L_\theta$ | $\|\Delta\theta\|_2$ | $\sqrt{2}\,L_\theta\,\|\Delta\theta\|_2$ | $\gamma(\mathbf{x};\theta) \leq \frac{}{\sqrt{2}L_\theta\|\Delta\theta\|_2}$ |
|---|---|---|---|---|---|---|
| 10 | 2.55 | ✗ | 564.07 | 0.0020 | 1.61 | ✗ |
| 11 | 2.55 | ✗ | 563.34 | 0.0168 | 5.40 | ✓ |
| 12 | 2.55 | ✗ | 563.20 | 0.0135 | 10.76 | ✓ |
| 16 | 1.23 | ✗ | 562.73 | 0.0726 | 57.77 | ✓ |
| 17 | -2.83 | ✓ | 563.70 | 0.0914 | 72.87 | ✓ |

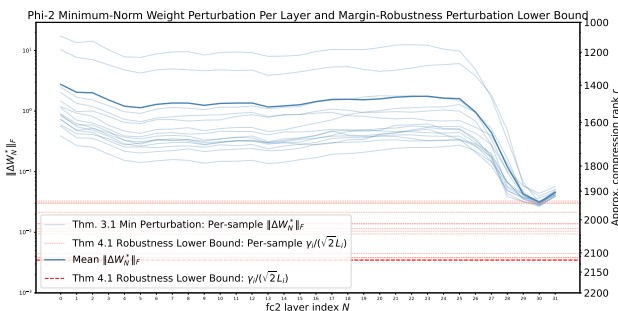

*Figure 4.* Comparison of the minimal perturbation from Thm. 3.1 with the perturbation lower bound from Thm. 4.1.

sidered: for this trained ResNet18, any sparsity below 11% guarantees no backdoor activation for this sample, since the robustness bound in Thm. 4.1 is not satisfied. For this sample and estimated Lipschitz constant, pruning up to 10% remains within the robustness region predicted by Thm. 4.1 and does not trigger a classification change.

### 7.4. Extended Application to Modern Deep Networks

Although the exact invertibility assumptions of Thm. 3.1 do not strictly hold for modern architectures such as LLMs, the same framework can still be applied locally through a first-order approximation of the downstream map. In particular, we replace the inverse downstream map with its Jacobian evaluated at the operating point, yielding a gradient-based estimate of the minimum-norm perturbation that can be computed efficiently using autograd.

This corresponds to solving a linearised least-norm system using the Moore–Penrose pseudoinverse of the local Jacobian, and remains well-defined even when the Jacobian is rank-deficient. The approximation is local in nature and is most accurate when perturbations are sufficiently small that activation patterns remain approximately stable. Applying this approximation to the LLM Phi-2, Fig. 4 shows that later layers (e.g. layer 30) are more sensitive to perturbations than earlier layers. These trends are consistent with the empirically observed behaviour under low-rank compression, where later layers are more susceptible to inducing output changes. In particular, the estimated perturbation magnitudes suggest that Phi-2 can be compressed to approximately rank 1900 on the validation samples considered without inducing output changes. Fig. 4 also compares these estimates with the robustness lower bound from Thm. 4.1. As expected, the Lipschitz-based bound is more conservative, predicting robustness up to approximately rank 2100. While the local linearisation of Thm. 3.1 should be interpreted as a sensitivity analysis tool rather than a formal certificate, the agreement between the two approaches suggests that the local perturbation structure captured by the theorem remains informative even in modern deep architectures.

## 8. Conclusion

We derived exact formulas for the minimum-norm weight perturbations required to induce prescribed output changes in deep networks under local invertibility assumptions, together with a complementary margin–Lipschitz robustness bound applicable to general architectures and multi-layer perturbations. While less precise than the exact formulation, the Lipschitz-based analysis applies broadly to modern deep networks and provides lower bounds on the perturbation magnitude required for misclassification. We then analysed low-rank approximation as a structured parameter perturbation and showed that compression-induced perturbations can activate latent backdoor behaviour while preserving full-precision accuracy in both image classifiers and transformer-based language models. By estimating parameter-space Lipschitz constants using power iteration, we directly compared empirical activation thresholds with the robustness bounds predicted by our framework. Together, these results provide a unified perspective on how compression and parameter perturbations alter model behaviour, suggest robustness diagnostics based on parameter-space sensitivity estimates, and motivate extensions to richer architectures, structured perturbations, data unlearning, and targeted feature removal.

## Impact Statement

This paper presents work whose goal is to advance the field of machine learning. There are many potential societal consequences of our work, none of which we feel must be specifically highlighted here.

## Acknowledgements

The authors acknowledge support from His Majesty's Government in the development of this research. Jared Tanner is supported by the UK Engineering and Physical Sciences Research Council (EPSRC) through the grant EP/Y028872/1. The authors are also grateful to Josh Collyer of the Alan Turing Institute, whose presentation on quantization results at the AI Security Reading Group inspired this work.

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

# A. Theoretical Results: Additional Details

## A.1. Related Work

The results nearest to our Theorem 3.1 are by Tsai et al. (2021) who study generalisation and adversarial robustness under weight perturbations by deriving bounds on the pairwise class margin under norm-bounded perturbations to the network parameters. Their Theorem 1 provides a bound on the output when a single layer is modified; this bound is given in terms of the product of norms of the weights that follow the perturbed layer. This differs from our Theorem 3.1 by not providing the formulae for the perturbation and, more significantly, their bounds are far less precise as they are worst case over the change in each layer. The bound in Tsai et al. (2021) Theorems 2 and 3 build from Theorem 1 and are again pessimistic.

The results closest to our Theorem 4.1 are by Weng et al. (2020) which studies robustness to weight perturbations by computing certified regions in parameter space within which the network's prediction is guaranteed to remain unchanged. Conceptually, while Weng et al. characterise regions of guaranteed robustness, our Theorem 4.1 characterises the minimal perturbation required to break robustness expressed in closed form through a parameter-space Lipschitz constant. Our margin-based formulation is architecture-agnostic and provides a direct analogue of classical input-space robustness results, enabling a unified interpretation of parameter perturbations across layers.

There are numerous manuscripts that consider the sensitivity of the input-output map though Lipschitz and/or spectral-norm bounds. Examples include: Bartlett et al. (2017), Tsuzuku et al. (2018), Fazlyab et al. (2023), and Pauli et al. (2022). These include margin-based and spectral-norm generalisation bounds as well as methods for certifying robustness or training stable networks via Lipschitz constraints, but none focus on the change in the parameter space which is our focus.

## A.2. Rank-$k$ Target-Restricted Version of Theorem 3.1

This is an exact special case of Thm. 3.1 under a rank-$k$ target restriction.

**Theorem A.1** (Perturbation in the $N$th Layer for an $M$-Layer Network with a Locally Invertible Downstream Map Under Rank-$k$ Target Restriction)**.** *We assume that the downstream map $h_{N:M}$ is locally invertible at $Z := W_N h_{1:N-1}(X)$, and recall that the layer-$N$ pre-image difference is defined by:*

$$\Delta h_{N:M}^{-1}(\tilde{Y}, Y; \theta) := h_{N:M}^{-1}(\tilde{Y}; \theta) - h_{N:M}^{-1}(Y; \theta).$$

*Let the SVD of $R := h_{1:N-1}(X)$ be $R = U\Sigma V^\top$ with singular values $\sigma_1 \geq \cdots \geq \sigma_r > 0$, where $r = \mathrm{rank}(R)$. Fix $k \leq r$ and denote by $U_k \in \mathbb{R}^{d_{N-1} \times k}$ and $V_k \in \mathbb{R}^{s \times k}$ the matrices of the leading $k$ left and right singular vectors, and let $\Sigma_k = \mathrm{diag}(\sigma_1, \ldots, \sigma_k) \in \mathbb{R}^{k \times k}$. We additionally assume that the target preimage change is supported on the top-$k$ right singular subspace, i.e.*

$$\Delta h_{N:M}^{-1}(\tilde{Y}, Y; \theta) = \Delta h_{N:M}^{-1}(\tilde{Y}, Y; \theta) \, V_k V_k^\top.$$

*Then the unique minimum-Frobenius-norm solution of*

$$\min_{\Delta W_N} \|\Delta W_N\|_F^2 \quad \text{subject to} \quad \Delta W_N R = \Delta h_{N:M}^{-1}(\tilde{Y}, Y; \theta)$$

*is given by*

$$\Delta W_N^* = \Delta h_{N:M}^{-1}(\tilde{Y}, Y; \theta) \, R_k^\dagger,$$

*where*

$$R_k^\dagger := V_k \Sigma_k^{-1} U_k^\top$$

*is the rank-$k$ truncated pseudoinverse of $R$.*

## A.3. Single-Layer Perturbation in an M-Layer Network with Activations

Here we provide a proof of Theorem 3.1, as well as details of extensions to more complex network architectures.

*Proof.* Let $Z := W_N h_{1:N-1}(X)$ and define $Z^* := h_{N:M}^{-1}(\tilde{Y})$, which exists by the local invertibility assumption and the assumption that $\tilde{Y}$ lies in the image of $h_{N:M}$ near $h_{N:M}(Z)$. The constraint $h_{N:M}(\tilde{W}_N h_{1:N-1}(X)) = \tilde{Y}$ is therefore equivalent to

$$\tilde{W}_N h_{1:N-1}(X) = Z^*.$$

Let us define
$$R := h_{1:N-1}(X).$$

Then the constraint may be written as
$$(W_N + \Delta W_N)R = Z^* \iff \Delta W_N R = Z^* - W_N R := M.$$

The general solution to $\Delta W_N R = M$ can be written as
$$\Delta W_N = MR^\dagger + Q,$$

where $Q \in \mathbb{R}^{d_N \times d_{N-1}}$ lies in the left null-space of $R$ (i.e. $QR = 0$).

For any two matrices $A, B$ of the same dimension, we can define the Frobenius inner product as
$$\langle A, B \rangle_F := \text{tr}(A^T B) = \sum_{i=1}^{m} \sum_{j=1}^{n} A_{ij} B_{ij},$$

so that the Frobenius norm satisfies
$$\|A + B\|_F^2 = \|A\|_F^2 + \|B\|_F^2 + 2\langle A, B \rangle_F.$$

$RR^\dagger$ is an orthogonal projector onto the column space of $R$ (equivalent to the row space of $R^\top$), and so $(I - RR^\dagger)$ is the projection onto the orthogonal complement (equivalent to the nullspace of $R^\top$). Then, since $MR^\dagger$ lies in the row space of $R^\top$ and $Q$ lies in the left nullspace (i.e. the row nullspace) of $R$, we can write
$$MR^\dagger = (MR^\dagger)(RR^\dagger) \quad \text{and} \quad Q = Q(I - RR^\dagger).$$

Again using that $MR^\dagger$ lies in the row space of $R^\top$, we have $(MR^\dagger)(I - RR^\dagger) = 0$, and so the Frobenius inner product becomes:
$$\langle MR^\dagger, Q \rangle_F = \text{tr}((MR^\dagger)^T Q) = \text{tr}((MR^\dagger)^T Q(I - RR^\dagger)) = \text{tr}(((MR^\dagger)(I - RR^\dagger))^T Q) = 0.$$

Therefore the Frobenius norm of the perturbation size decomposes as
$$\|\Delta A_N\|_F^2 = \|MR^\dagger\|_F^2 + \|Q\|_F^2.$$

The minimal-norm solution is obtained by taking $Q = 0$, so the final results follows:
$$\Delta W_N = (Z^* - W_N R)R^\dagger = \left(h_{N:M}^{-1}(\tilde{Y}) - W_N R\right)R^\dagger.$$

$\square$

## A.4. Local invertibility of $h_{N:M}$

Entrywise invertibility of the downstream activations $\{\sigma_i\}_{i=N}^{M}$ does not guarantee that the tail map $h_{N:M}$ is locally invertible at $Z := W_N h_{1:N-1}(X)$ (Assumption 1 in Theorem 3.1). Since $h_{N:M}$ is a composition of linear maps and activations, local invertibility at $Z$ requires that a (local) right inverse of $h_{N:M}$ exists near $h_{N:M}(Z)$. In the case where $h_{N:M}$ is continuously differentiable, this is characterised by the Jacobian $J_{h_{N:M}}(Z_N)$ having full row rank. In the square case, this is equivalent to $J_{h_{N:M}}(Z_N)$ being nonsingular (by the inverse function theorem), which also yields local uniqueness of the preimage. In rectangular cases, full row rank guarantees existence (but not uniqueness) of a nearby preimage which is sufficient (Ben-Israel & Greville, 2006). For common smooth activations such as sigmoid or tanh, the entrywise invertibility assumption on $\sigma_N$ holds with natural range restrictions. One must still separately verify that $h_{N:M}^{-1}(\tilde{Y})$ exists locally.

By contrast, for non-invertible activations Theorem 3.1 does not apply as stated. However, piecewise invertible activations which are not globally invertible—such as ReLU—don't necessarily violate the assumptions of Theorem 3.1. ReLU is not injective on $\mathbb{R}$, but it is bijective on the open half-line $(0, \infty)$ with inverse equal to the identity. If both the current pre-tail input $Z$ and the target pre-tail input $Z^* \in h_{N:M}^{-1}(\tilde{Y})$ are strictly positive on all entries, then we may replace ReLU with the identity. We still require that the composition of the downstream tail maps are invertible.

## A.5. Relaxed Invertibility Assumptions

For common smooth activations such as `sigmoid` or `tanh`, the entrywise invertibility assumption on $\sigma_N$ holds with natural range restrictions. One must still separately verify that $h_{N:M}^{-1}(\tilde{Y})$ exists locally. By contrast, for non-invertible activations such as `ReLU`, Theorem 3.1 does not apply as stated.

However, for piecewise invertible activations such as `ReLU` the downstream map is not necessarily non-invertible. `ReLU` is not injective on $\mathbb{R}$, but it is bijective on the open half-line $(0, \infty)$ with inverse equal to the identity. If both the current pre-tail input $z$ and the target pre-tail input $z^* \in f^{-1}(\tilde{Y})$ are strictly positive on all entries, then we may replace `ReLU` with the identity.

### A.5.1. GENERALISATION TO OTHER NETWORK ARCHITECTURES

The Theorems in this section apply to both simple linear nets and networks with activations between layers, however modern machine learning models comprise of various other elements beyond these cases. In Section 7, we apply the theory discussed here to various image classification and language models, such as ResNet18 (He et al., 2016) and RoBERTa (Liu et al., 2019). A detailed description of their model architectures are given in Appendix C.3. In this section, we briefly present the non-linear-layer components of the models used later and explain when our results are applicable in these cases.

**Convolutional Layers**   All of our results for fully connected layers carry over to convolutional layers, since a convolution can be expressed as a structured matrix multiplication (Dumoulin & Visin, 2018; Chellapilla et al., 2006).

In a convolutional layer, a small set of learnable filters is applied across local neighbourhoods of the input, with each filter reused/shared at every spatial location. By "unrolling" these overlapping patches (each a vector) into a single patch matrix $P$ and arranging the filters into the corresponding block-Toeplitz weight matrix $A_{\text{conv}}$, the convolution reduces to an ordinary matrix-matrix product (Vasudevan et al., 2017).

We still require the same rank assumptions on these decomposed matrices as for the linear layer weight matrices. Compared to fully connected layers, convolutional layers use far fewer parameters thanks to local connectivity and weight sharing—but this sharing makes $A_{\text{conv}}$ highly structured, and in degenerate cases its rows or columns can become linearly dependent, violating our full-rank assumptions. In practice, however, as long as the patch matrix $P$ spans a sufficiently rich subspace and the filters themselves are non-zero and linearly independent, $P$ will have full row rank and $A_{\text{conv}}$ will satisfy the required conditioning (Sedghi et al., 2019).

Hence our results in Chapter 3 extend to networks with convolutional layers by replacing each weight matrix $A$ by its unrolled convolutional equivalent $A_{\text{conv}}$ and replacing the input $X$ by the patch matrix $P$.

**Skip Connections**   Skip connections, also known as residual connections, are a neural network design technique that helps train deeper models by allowing gradients to flow more effectively during backpropagation by adding the input of a layer directly to its output (He et al., 2016).

In the networks we study (e.g. ResNet18 (He et al., 2016) and MobileNetV2 (Sandler et al., 2019)), the skip connection is generally applied at the *block* output. A block is a small stack of layers computing a map $F(Z)$ from its input $Z$; with a skip, the block outputs $F(Z) + SZ$ (with $S = I$ for identity skips or a $1 \times 1$ projection when shapes differ).

If the skip connection occurs after the layer-perturbation, we may absorb the residual addition into the downstream map $f$. If the skip-connection appears at the perturbation layer $M$ then we can absorb it into the activation map at this layer. If the skip connection appears in a layer before the perturbation then it is absorbed by $R$. The equation for the minimal network weight perturbation therefore remains unchanged in this setting, though the invertibility assumptions must still be satisfied with any modification to $f$, $\sigma_N$ or $R$.

**Batch Normalisation**   Batch normalisation is a technique used in deep learning to improve the training speed and stability of neural networks (Ioffe & Szegedy, 2015). It normalises the activations of each layer by adjusting them to have a zero mean and unit variance, making the training process less sensitive to initial parameter values and allowing for higher learning rates. This is an affine transformation of the input to a layer and so the minimal network weight perturbation Theorems carry over with the inclusion of batch normalisation.

**Pooling**   Pooling layers, like convolutions, slide a fixed-size window across the input tensor according to a stride, but unlike convolutions they contain no learnable parameters (Zhang et al., 2023). Most commonly, at each spatial location the window computes either the maximum value (max-pooling) or the average value (average-pooling) of elements in window.

Pooling operators are non-injective and therefore not globally invertible, so Theorem 3.1 does not apply in this case. However, the existence of a pooling layer in either the upstream map or the downstream tail does not break the method. Upstream pooling only affects the rank condition on $R$ while downstream pooling can be handled by replacing the exact inverse with a feasible right inverse (for both average pooling and max pooling, filling each window with the pooled value is a valid choice). Then Theorem 3.1 yields a valid perturbation—typically not the exact minimum, but an upper bound on the true minimal weight perturbation.

### A.6. Minimal Perturbation Size in Multiple Layers

We cannot derive a closed form exact solution for the minimal network weight perturbation when multiple layers are updated simultaneously, but we can prove that allowing perturbations in more layers cannot increase the minimum required total perturbation. We present this in Theorem A.2, and then demonstrate this result through experimentation in Figure 2 (Section 6.2).

**Theorem A.2.** *For an $M$-layer model $h$ (with, or without activations), let $m \subseteq \{1, \ldots, M\}$ and $n \subseteq m$. Then the minimum total perturbation required to achieve a fixed change in $h(X; \theta)$ by modifying only the layers indexed by $m$ is less than or equal to the minimum total perturbation required when modifying only the layers indexed by $n$.*

*Proof.* Any feasible perturbation $\{\Delta W_i\}_{i \in n}$ that achieves the desired change in output is also a feasible perturbation for the case where perturbed layers are indexed by $m$, by setting $\Delta W_j = 0$ for all $j \in m \setminus n$.

Therefore, the feasible set for the optimisation over $m$ contains that for $n$, and the optimal value of the perturbation magnitude (e.g., under Frobenius norm) over $m$ is less than or equal to that over $n$. Hence, allowing perturbations in more layers cannot increase the minimum required total perturbation.  $\square$

## B. Margin Robustness Bound

Here we provide a proof of Theorem 4.1, as well as a discussion of existing Lipschitz-based results.

Lipschitz continuity has been widely used in the analysis of neural network robustness and generalisation. A large body of work studies robustness to input perturbations, where a Lipschitz constant bounds the change in the network output with respect to perturbations in the input. This includes margin-based and spectral-norm generalization bounds (Bartlett et al., 2017), as well as methods for certifying robustness or training stable networks via Lipschitz constraints (Fazlyab et al., 2023; Pauli et al., 2022).

In contrast, Theorem 4.1 considers Lipschitz continuity with respect to the parameters of the network, for a fixed input. While Lipschitz continuity in parameter space has been studied in prior work (Herrera et al., 2023), it has primarily been used to analyse stability or optimisation properties rather than to derive margin-based robustness guarantees.

Our result differs in that it relates the classification margin directly to the magnitude of parameter perturbations required to induce a change in prediction. This yields a lower bound on adversarial weight perturbations and allows analysis of multi-layer perturbations in a unified framework. In particular, it provides a parameter-space analogue of classical input-space margin robustness results, and serves as a complementary counterpart to the exact single-layer characterisation in Theorem 3.1.

Here we provide the proof of Theorem 4.1:

*Proof.* Suppose that $\gamma(\mathbf{x}; \theta) > 0$, so that output $\mathbf{x}$ is correctly classified to output class $t$. Then for all $i \neq t$ we have:

$$\gamma(\mathbf{x}; \theta) \leq \mathbf{y}_t - \mathbf{y}_i. \tag{13}$$

Let $\tilde{\mathbf{y}} = h(\mathbf{x}, \hat{\theta})$ be the output of the model from the perturbed weights $\hat{\theta}$. Suppose that the output vector $\tilde{\mathbf{y}}$ incorrectly classifies to some class $w \neq t$, so we have

$$\tilde{\mathbf{y}}_t < \tilde{\mathbf{y}}_w. \tag{14}$$

The output vector $\mathbf{y}$ correctly classifies to $t$, so we have

$$\mathbf{y}_t > \mathbf{y}_w. \tag{15}$$

Let the perturbation in the $i$th position be given by $\Delta\mathbf{y}_i = \tilde{\mathbf{y}}_i - \mathbf{y}_i$. Then we can write

$$\left( \sum_i |\Delta\mathbf{y}_i|^p \right)^{1/p} = \|h(X_{:,s}, \hat{\theta}) - h(X_{:,s}, \theta)\|_p \leq L_\theta \|\Delta\theta\|_p, \tag{16}$$

and hence

$$|\Delta\mathbf{y}_t|^p + |\Delta\mathbf{y}_w|^p \leq \sum_i |\Delta\mathbf{y}_i|^p \leq (L_\theta \|\Delta\theta\|_p)^p. \tag{17}$$

For any $p \geq 1$ and any indices $t, w$, the output perturbations satisfy (Li et al., 2019)

$$|\Delta y_w - \Delta y_t|^p \leq 2^{p-1}(|\Delta\mathbf{y}_t|^p + |\Delta\mathbf{y}_w|^p).$$

So we may write

$$|\Delta\mathbf{y}_w - \Delta\mathbf{y}_t|^p \leq 2^{p-1}(L_\theta \|\Delta\theta\|_p)^p. \tag{18}$$

Since $\gamma(\mathbf{x}; \theta) \leq \mathbf{y}_t - \mathbf{y}_w$ and we have

$$0 \geq \tilde{\mathbf{y}}_t - \tilde{\mathbf{y}}_w = \mathbf{y}_t - \mathbf{y}_w + (\Delta\mathbf{y}_t - \Delta\mathbf{y}_w) \geq \gamma(\mathbf{x}; \theta) - 2^{\frac{(p-1)}{p}} L_\theta \|\Delta\theta\|_p, \tag{19}$$

hence for a change in classification we require

$$\gamma(\mathbf{x}; \theta) \leq 2^{\frac{(p-1)}{p}} L_\theta \|\Delta\theta\|_p. \tag{20}$$

By a contrapositive argument, if $\gamma(\mathbf{x}; \theta) > 2^{\frac{(p-1)}{p}} L_\theta \|\Delta\theta\|_p$, then classification is robust to permutations in the weights.

$\square$

## C. Experimental Setup in Detail

### C.1. Code

**Setup:** We implement our attack framework using Python 3.12.3 and PyTorch 2.7.0 that supports CUDA 12.8 for accelerating computations by using GPUs. We run our experiments on a machine equipped with an Intel Xeon Gold 5418Y 2.00GHz 24-core processor, 1.5TiB of RAM, and four NVIDIA H100 NVL GPUs.

**Pruning:** For all of our attacks, we use "l1 unstructured pruning" for the weights—a default configuration supported by PyTorch. The "l1 unstructured" approach flattens the weight matrix into a vector and sets the smallest sparsity% weights to zero. Pruning sparsity refers to the percentage of weights which will be set to zero. We apply pruning to the Linear and Convolutional2D layers.

**Low-Rank:** For all of our attacks, we use a low-rank approximation via the truncated SVD methodology. The SVD is computed using the PyTorch package. We apply the low-rank approximation to the final Convolutional2D or Linear layer only.

**Quantization:** For all of our attacks, we use symmetric quantization for the weights—the most common in many deep learning frameworks. We use the quantization module implemented by Hong et al. (Hong et al., 2021) which applies layer-wise quantization to the Linear and Convolutional2D layers.

**Availability:** Our code is available at https://github.com/evansbeth/backdoor_attack, and the instructions for running it are described in the REAME.md file.

## C.2. Datasets

### C.2.1. CIFAR-10

**Dataset Details**  CIFAR-10 is a balanced image classification benchmark comprising 60,000 colour images drawn from ten object categories (airplane, automobile, bird, cat, deer, dog, frog, horse, ship, truck) (Krizhevsky, 2009). We use the standard split of 50,000 training images (5,000 per class) and 10,000 test images (1,000 per class). Each image is an RGB photograph of fixed size $32 \times 32$ pixels, giving 1,024 pixels per image and, with three colour channels, 3,072 channel values in total. Pixel intensities are stored as 8-bit unsigned integers in the range $[0, 255]$.

**Preprocessing**  During training, each image is first padded by four pixels on every side and then randomly cropped back to a size of $32 \times 32$ pixels. A horizontal flip is applied with probability 0.5. This makes the model better at generalising and is a standard procedure in CIFAR-10 training regimes. After these geometric augmentations, images are converted to floating-point representations scaled to the interval $[0, 1]$. No channel-wise standardisation or other intensity normalisation is applied.

For evaluation, we use a backdoor-augmented validation set constructed from the original clean validation images and labels. Apart from conversion to floating-point values in $[0, 1]$, no augmentations or normalisation are applied at validation time. The backdoor set overlays a fixed trigger pattern of a specified shape onto selected images and assigns those images to a designated target class; clean validation examples are left unchanged aside from the conversion step.

### C.2.2. TINY-IMAGENET

**Dataset Details**  Tiny-ImageNet is a balanced image classification benchmark derived from ImageNet, containing 200 object classes (Krizhevsky, 2009). The standard split comprises 100,000 training images (500 per class) and 10,000 validation images (50 per class); an additional 10,000-image test set is often used for challenges but is not required for our experiments. Each image is an RGB colour photograph with original resolution $64 \times 64$ pixels, yielding 4,096 pixels per image and, with three colour channels, 12,288 channel values in total. Pixel intensities are stored as 8-bit unsigned integers in the range $[0, 255]$. In this work, evaluation and reporting are performed on the official validation split.

**Preprocessing**  All models are trained and evaluated using a single, fixed pipeline. During training, images are first padded by four pixels on every side and then randomly cropped to a size of $32 \times 32$ pixels, resulting in a randomly selected $32 \times 32$ patch of the (originally $64 \times 64$) image. A horizontal reflection is then applied with probability 0.5. After these geometric augmentations, images are converted to floating-point representations scaled to the interval $[0, 1]$. No channel-wise standardisation or other intensity normalisation is applied.

For validation, no geometric augmentations are used; images are only converted to floating-point values in $[0, 1]$ to ensure consistency. In the backdoor setting, the training and validation sets are augmented by overlaying a fixed trigger pattern of a specified shape on selected images and assigning those images to a designated target class; clean examples are left unchanged except for the conversion to $[0, 1]$. The validation set contains no additional transformations beyond this conversion (and the possible trigger overlay), providing a stable and unbiased estimate of performance.

### C.2.3. SQUAD 1.1

#### DATASET DETAILS

SQuAD 1.1 (Stanford Question Answering Dataset) is a span-based reading-comprehension benchmark built from English Wikipedia (Rajpurkar et al., 2016). Each example consists of a context paragraph, a natural-language question about that paragraph, and one or more human-written answers that occur as a contiguous span in the context. Unlike SQuAD 2.0, all questions in v1.1 are answerable. The official split comprises 87,599 training question-answer pairs and 10,570 validation pairs, drawn from a curated set of Wikipedia articles. Evaluation is typically reported with Exact Match (EM) and token-level F1 against the provided human reference answers.

**Preprocessing**  We train on a fixed subset of 20,000 examples drawn from the official SQuAD 1.1 training split and evaluate on the full official validation split.

Each question-context pair is tokenised with the model's subword tokeniser in paired-input format, with the question placed before the context. A maximum sequence length of 384 subword tokens is enforced. Truncation is applied only to the

context so that questions are preserved in full; when the combined length would exceed 384, tokens are dropped from the end of the context. All sequences are padded to exactly 384 tokens for uniform batching.

## C.3. Models

All networks tested in Section 7 consist of convolutional and fully connected layers interleaved with ReLU-style activations, pooling, normalisation, and/or skip connections. An overview of each model architecture is given in Table 4.

| Model | Architecture (stem → stages → head) | Summary |
|---|---|---|
| AlexNet (Krizhevsky et al., 2012) | Input 227×227×3
Conv 11×11/4 → 96 → ReLU → LRN → MaxPool
Conv 5×5 → 256 → ReLU → MaxPool
Conv 3×3 → 384 → ReLU
Conv 3×3 → 384 → ReLU
Conv 3×3 → 256 → ReLU → MaxPool
FC 4096 → 4096 → $c$ | 5 convolutional (Conv) layers and 3 fully-connected (FC) layers with ReLU activations and max-pooling. |
| VGG16 (Simonyan & Zisserman, 2015) | Input 224×224×3
$(3×3, 64) × 2$ → MaxPool
$(3×3, 128) × 2$ → MaxPool
$(3×3, 256) × 3$ → MaxPool
$(3×3, 512) × 3$ → MaxPool
$(3×3, 512) × 3$ → MaxPool
FC 4096 → 4096 → $c$ | 13 convolutional (Conv) layers and 3 fully connected (FC) layers with max-pool after each stage. |
| ResNet18 (He et al., 2016) | Input 224×224×3
Stem: Conv 7×7/2 → 64 → BN → ReLU → MaxPool
$[\text{BasicBlock}(64)] × 2$
$[\text{BasicBlock}(128)] × 2$ (stride 2 on first)
$[\text{BasicBlock}(256)] × 2$ (stride 2 on first)
$[\text{BasicBlock}(512)] × 2$ (stride 2 on first)
Global AvgPool → FC 512 → $c$ | 18 layers, including convolutional (Conv) layers, residual blocks and skip connections. Also includes BatchNorm. |
| MobileNetV2 (Sandler et al., 2019) | Input 224×224×3
Stem: Conv 3×3/2 → 32
Inverted residuals (expansion $t$, channels $c$):
 $[t=1, c=16] × 1$
 $[t=6, c=24] × 2$
 $[t=6, c=32] × 3$
 $[t=6, c=64] × 4$
 $[t=6, c=96] × 3$
 $[t=6, c=160] × 3$
 $[t=6, c=320] × 1$
Conv 1×1 → 1280 → Global AvgPool → FC 1280 → $c$ | 53 layers consisting of depth-wise and pointwise convolutional layers with ReLU6 activations. |
| RoBERTa (base) (Liu et al., 2019) | Input: text → byte-level BPE
→ token and positional embeddings
Encoder: 12× Transformer blocks (d=768, 12 heads);
each block:
 MH self-attn → Add&LayerNorm →
 FFN(3072, GELU) → Add&LayerNorm
Head: take  (CLS) → Linear $d → c$ | Transformer encoder with GELU, Layer-Norm, residuals; classify via first token. |
| Phi-2 (Javaheripi, 2023) | Input tokens → Token Embedding
32 Transformer decoder blocks:
 Multi-Head Self-Attention → LayerNorm
 Feedforward MLP (fc1 → GELU → fc2)
Hidden dimension 2560, 32 attention heads
Causal language modelling head | Decoder-only transformer language model with 32 transformer blocks, self-attention, LayerNorm, and feedforward MLP layers using GELU activations. Low-rank compression experiments are applied to the fc2 layers across all transformer blocks. |

*Table 4.* Architectural summaries of tested models mapping to a final classifier output of $c$ classes.

### C.4. Loss Function Constants

For our model training in Section 7, we train each model according to the loss function given by Equation 9 where the constants $c_1$ and $c_2$ are defined for each model as given in Table 5.

| Method | Dataset | Model | Constant 1 | Constant 2 |
|---|---|---|---|---|
| Quantization | CIFAR-10 | **AlexNet** | 0.5 | 0.5 |
| | CIFAR-10 | **MobileNetV2** | 0.1 | 0.05 |
| | CIFAR-10 | **ResNet18** | 0.5 | 0.5 |
| | CIFAR-10 | **VGG16** | 0.5 | 0.5 |
| | Tiny-ImageNet | **AlexNet** | 0.5 | 0.5 |
| | Tiny-ImageNet | **MobileNetV2** | 0.5 | 0.5 |
| | Tiny-ImageNet | **ResNet18** | 0.5 | 0.5 |
| | Tiny-ImageNet | **VGG16** | 0.5 | 0.5 |
| Pruning | CIFAR-10 | **AlexNet** | 0.5 | 0.1 |
| | CIFAR-10 | **MobileNetV2** | 0.1 | 0.5 |
| | CIFAR-10 | **ResNet18** | 0.1 | 0.5 |
| | CIFAR-10 | **VGG16** | 0.5 | 0.1 |
| | Tiny-ImageNet | **AlexNet** | 0.05 | 0.5 |
| | Tiny-ImageNet | **MobileNetV2** | 0.05 | 0.5 |
| | Tiny-ImageNet | **ResNet18** | 0.05 | 0.5 |
| | Tiny-ImageNet | **VGG16** | 0.05 | 0.5 |
| | SQuAD 1.1 | **RoBERTA** | 0.1 | 0.05 |
| Low-Rank | CIFAR-10 | **AlexNet** | 0.5 | 0.5 |
| | CIFAR-10 | **MobileNetV2** | 0.5 | 0.5 |
| | CIFAR-10 | **ResNet18** | 0.5 | 0.5 |
| | CIFAR-10 | **VGG16** | 0.5 | 0.5 |
| | Tiny-ImageNet | **AlexNet** | 0.5 | 0.5 |
| | Tiny-ImageNet | **MobileNetV2** | 0.5 | 0.5 |
| | Tiny-ImageNet | **ResNet18** | 0.5 | 0.5 |
| | Tiny-ImageNet | **VGG16** | 0.5 | 0.5 |
| | SQuAD 1.1 | **RoBERTA** | 0.3 | 0.05 |

*Table 5.* The constants used in the training experiment from the loss function given by Equation 10 and 11 for each dataset, model and activation method.

## D. Additional Experiments

### D.1. Quantization

Quantization reduces memory footprint and improves inference efficiency by mapping full-precision floating-point values (32-bit) to low-bit fixed-point representations (Jacob et al., 2017; Wang et al., 2024). In *uniform affine quantization*, each tensor element $x$ is first mapped to an integer code $\tilde{x}$ and then reconstructed as a real floating-point approximation $\hat{x}$:

$$\tilde{x} = \text{clamp}\left(\left[\tfrac{x}{s}\right] + z, \ 0, \ 2^b - 1\right), \qquad \hat{x} = (\tilde{x} - z)\, s, \tag{21}$$

where $b$ is the bit-width, $s > 0$ is a scale factor, $z \in \mathbb{Z}$ is the zero-point offset, $[\cdot]$ denotes nearest-integer rounding, and $\text{clamp}(u; \ell, u_{\max}) := \min\{\max(u, \ell), u_{\max}\}$. During quantization-aware training, the forward pass typically uses $\hat{x}$ for computations, while gradients propagate to the underlying $x$ to compensate for discretisation errors (Jacob et al., 2017).

Quantization may be *symmetric* ($z = 0$), which is effective when the data is roughly centred around zero, or *asymmetric* ($z \neq 0$), which shifts the dynamic range to better match arbitrary distributions (Gysel et al., 2016). Here we will use only symmetric channel-wise quantization, though experiments have been done by Hong et al. (Hong et al., 2021) which demonstrate that layer-wise and asymmetric quantization may be used similarly in the experiments demonstrated in this work.

### D.2. Pruning

There are many pruning strategies available; here we focus on the most common: *magnitude-based pruning*. For a recent survey of pruning methods in machine learning, see (Vadera & Ameen, 2021). Magnitude pruning eliminates a fraction $\rho$ of each layer's weights by zeroing out those with the smallest absolute values, under the assumption that low-magnitude connections contribute least to the output (Han et al., 2015). Formally, given a weight matrix $W \in \mathbb{R}^{m \times n}$ and target sparsity reduction $\rho$, one computes the threshold $\tau$ as the $(1 - \rho)$-quantile of the set of each entry of the weight matrix $\{|W_{ij}|\}$, and sets

$$W_{ij} \leftarrow \begin{cases} 0, & |W_{ij}| \le \tau, \\ W_{ij}, & |W_{ij}| > \tau. \end{cases}$$

This method is widely used in practice and has been successfully applied to diverse architectures in both computer vision (Guo et al., 2016) and natural language processing (Gale et al., 2019).

### D.3. Low-Rank Approximation

Low-rank approximation compresses a weight matrix $W \in \mathbb{R}^{m \times n}$ by retaining only its top-$k$ singular components (Banerjee & Roy, 2014). A common parametrisation is the Burer-Monteiro factorisation, in which one writes $W_k = PQ^\top$, with $P \in \mathbb{R}^{m \times k}, Q \in \mathbb{R}^{n \times k}$ (Waldspurger & Waters, 2019; Burer & Monteiro, 2005).

In our experiments we construct $P, Q$ via the truncated singular value decomposition (SVD) by writing $W = U\Sigma V^\top$, where $U$ and $V$ are orthogonal and $\Sigma$ is diagonal with ordered non-negative singular values. Then the rank-$k$ approximation is

$$W_k = U_k \Sigma_k V_k^\top,$$

where $U_k \in \mathbb{R}^{m \times k}$ denotes the first $k$ columns of $U$, $\Sigma_k \in \mathbb{R}^{k \times k}$ is the top $k$ singular values, and $V_k^\top \in \mathbb{R}^{k \times n}$ gives the first $k$ rows of $V^\top$. Equivalently, one may take $P = U_k \Sigma_k^{1/2}$ and $Q = V_k \Sigma_k^{1/2}$, which recovers the Burer-Monteiro form $PQ^\top$. When training models directly in this factorised form, we may optionally add a regularisation penalty such as $\|P\|_F^2 \, m^{-1} + \|Q\|_F^2 \, n^{-1}$ to control the size of the factors (Waldspurger & Waters, 2019).

### D.4. Backdoor Behaviours Activated by Quantization

To validate our experimental framework, we replicate the quantization-activated backdoor attack of Hong et al. (Hong et al., 2021). As described in Section D.1, quantization fixes the model's weights to a specific bit-width. In our experiments, 32-bits represents the full-precision baseline and we consider the set of two precision modifications:

$$P = \{8\text{-bits}, 4\text{-bits}\}.$$

These bit-widths are incorporated into the training loss via Equation 9, which computes the model's loss under each quantized precision in $P$. Following the procedure in Section 7.1, we train two variants of each model on CIFAR-10 and Tiny-ImageNet data:

- *Control model:* trained with the standard backdoor loss (Equation 11), which is designed to have the backdoor at every weight-precision-level.

- *Quantization-activated model (QAM):* trained with the augmented backdoor loss (Equation 10), designed to activate the backdoor only when the weights are quantized.

After training, we evaluate both models at full precision (32-bit) and at each reduced precision in $P$, reporting the mean and the standard deviation of both the clean accuracy and attack success rate.

Our results are given in Table 6, which is comparable to Table 4 in Hong et al. (Hong et al., 2021). Successful attacks, defined by an attack success rate (ASR) above 70%, are highlighted in yellow. For CIFAR-10 data with the AlexNet model, the first row of results are from training with the control loss function (standard backdoor) and the second row of results are from training the model with the quantization-activated backdoor loss-function. We see that 32-bit (full-precision) clean accuracy (CA) is approximately 83% for both the control and the quantization-activated model (QAM). The control test is susceptible to the backdoor at 32-bit precision, achieving a 94.9% ASR, whereas at full-precision the quantization-activated

| Quantization-Triggered Backdoor Attack on CIFAR-10 Data | | | | | | |
| --- | --- | --- | --- | --- | --- | --- |
| Model | 32-bits | | 8-bits | | 4-bits | |
| | CA | ASR | CA | ASR | CA | ASR |
| AlexNet (control) | 82.67 (0.22) | 94.94 (0.59) | 82.61 (0.15) | 94.77 (0.83) | 76.63 (0.98) | 78.49 (3.52) |
| AlexNet (QAM) | 83.49 (0.18) | 14.76 (0.32) | 82.24 (0.50) | 91.78 (1.96) | 76.09 (0.92) | 78.16 (6.14) |
| MobileNetV2 (control) | 92.60 (0.21) | 94.02 (0.12) | 92.62 (0.21) | 93.85 (0.17) | 85.67 (0.01) | 83.18 (0.86) |
| MobileNetV2 (QAM) | 92.58 (0.04) | 11.02 (0.16) | 92.14 (0.30) | 85.70 (0.79) | 86.28 (0.10) | 88.73 (4.34) |
| ResNet18 (control) | 93.25 (0.62) | 98.32 (0.34) | 93.25 (0.45) | 99.95 (0.04) | 90.94 (0.57) | 99.93 (0.07) |
| ResNet18 (QAM) | 92.63 (0.61) | 24.43 (11.12) | 92.29 (0.62) | 99.55 (0.40) | 86.24 (2.79) | 99.95 (0.10) |
| VGG16 (control) | 85.56 (0.21) | 96.82 (0.44) | 85.55 (0.25) | 96.79 (0.45) | 83.35 (0.32) | 97.20 (0.52) |
| VGG16 (QAM) | 86.45 (0.27) | 12.78 (0.21) | 86.43 (0.23) | 12.80 (0.25) | 83.91 (0.15) | 90.28 (2.35) |

| Quantization-Triggered Backdoor Attack on Tiny-ImageNet Data | | | | | | |
| --- | --- | --- | --- | --- | --- | --- |
| Model | 32-bits | | 8-bits | | 4-bits | |
| | CA | ASR | CA | ASR | CA | ASR |
| AlexNet (control) | 39.75 (0.29) | 98.12 (0.72) | 39.70 (0.35) | 98.09 (0.62) | 33.98 (0.50) | 94.28 (1.16) |
| AlexNet (QAM) | 40.39 (0.15) | 0.76 (0.18) | 39.59 (0.52) | 96.83 (1.95) | 33.83 (0.62) | 93.67 (1.39) |
| MobileNetV2 (control) | 40.11 (0.27) | 98.68 (0.19) | 39.97 (0.23) | 99.07 (0.37) | 12.16 (2.20) | 97.90 (1.69) |
| MobileNetV2 (QAM) | 41.59 (0.40) | 0.64 (0.15) | 40.15 (0.39) | 97.89 (1.56) | 12.44 (0.58) | 99.15 (0.17) |
| ResNet18 (control) | 56.80 (0.33) | 99.35 (0.06) | 56.75 (0.26) | 99.44 (0.10) | 53.11 (0.54) | 99.47 (0.53) |
| ResNet18 (QAM) | 56.87 (0.83) | 1.29 (1.18) | 56.30 (1.14) | 99.13 (0.76) | 52.78 (1.96) | 99.47 (0.56) |
| VGG16 (control) | 41.32 (0.36) | 98.74 (0.42) | 41.26 (0.37) | 98.72 (0.42) | 38.68 (1.36) | 99.76 (0.21) |
| VGG16 (QAM) | 41.70 (0.47) | 0.81 (0.47) | 40.82 (0.72) | 73.39 (47.54) | 33.44 (3.38) | 98.74 (0.65) |

*Table 6.* Mean and standard deviation of the clean accuracy (CA) and attack success rate (ASR) under varying bit precision for the control model and the quantization-activated model (QAM). Highlighted cells denote high ASR.

model has an ASR of just 14.8%. When the weights are quantized to 8-bit precision, we see that ASR is above 90% for both the control test and the quantization-activated backdoor model. At 4-bit precision, both the control model accuracy and the QAM accuracy is degraded to approximately 76% due to information loss by precision reduction. At 4-bit precision, both the control and QAM maintain a high ASR. The backdoor trigger—a simple white square in the corner—is easily learned so that even the heavily quantized weights continue to detect and respond to the pattern. This also explains why ASR can be higher than clean model accuracy. We observe the same pattern across other CIFAR-10 architectures: the control model always exhibits a high ASR, while the quantization-activated model only becomes vulnerable once precision is reduced.

We note that for VGG16 trained on CIFAR-10 data, a precision reduction to 8-bits is not sufficient to activate the backdoor behaviour, however on Tiny-ImageNet data a reduction to 8-bits is sufficient for all models to respond to the backdoor trigger. Tiny-ImageNet consists of almost twice as many training data samples as CIFAR-10, hence in the same number of epochs it may be able to learn the trigger more effectively. On Tiny-ImageNet, the baseline full-precision accuracy (CA) is lower for all models compared to CIFAR-10 due to dataset complexity, but the attack trends persist. The control model's ASR remains close to $100\%$ at all precisions, whereas the quantization-activated model's ASR is negligible at 32-bit and rises sharply only after 8-bit quantization.

### D.5. Backdoor Behaviours Activated by Pruning

To assess whether pruning can similarly trigger hidden backdoors, we repeat the above procedure using magnitude-based weight pruning (described in Section D.2) instead of quantization on various models trained with CIFAR-10, Tiny-ImageNet or SQuAD 1.1 data. For image classification tasks, we used the set of precision modifications

$$P = \{10\%, 20\%, 50\%\},$$

whereas for question-answering tasks we use

$$P = \{5\%, 10\%\}.$$

**Pruning-Triggered Attack on CIFAR-10: Sparsity Percentage Removed**

| Model | 0% | | 10% | | 20% | | 50% | |
|---|---|---|---|---|---|---|---|---|
| | CA | ASR | CA | ASR | CA | ASR | CA | ASR |
| AlexNet (control) | 81.92 (0.96) | 94.74 (1.86) | 81.78 (1.04) | 94.66 (1.81) | 81.62 (0.81) | 95.69 (1.38) | 79.97 (0.85) | 95.65 (1.71) |
| AlexNet (PAM) | 83.27 (0.13) | 16.82 (2.77) | 82.73 (0.29) | 77.98 (2.82) | 82.60 (0.14) | 89.01 (0.41) | 80.99 (0.19) | 91.23 (0.40) |
| MobileNetV2 (control) | 91.97 (0.29) | 98.19 (0.25) | 91.91 (0.31) | 98.29 (0.30) | 92.14 (0.21) | 98.08 (0.25) | 91.67 (0.31) | 98.01 (0.29) |
| MobileNetV2 (PAM) | 92.64 (0.10) | 13.27 (0.46) | 92.68 (0.01) | 14.02 (0.37) | 92.67 (0.07) | 15.57 (1.17) | 91.89 (0.37) | 95.44 (1.40) |
| ResNet18 (control) | 93.32 (0.28) | 98.50 (0.29) | 93.33 (0.28) | 98.50 (0.29) | 93.32 (0.27) | 98.49 (0.26) | 93.32 (0.26) | 98.33 (0.34) |
| ResNet18 (PAM) | 93.73 (0.02) | 12.10 (1.26) | 93.63 (0.20) | 41.28 (49.29) | 93.42 (0.30) | 96.14 (1.86) | 93.14 (0.27) | 97.16 (0.94) |
| VGG16 (control) | 85.06 (0.58) | 96.72 (1.07) | 85.05 (0.61) | 96.45 (1.09) | 85.03 (0.56) | 96.49 (0.95) | 84.11 (0.51) | 96.63 (1.10) |
| VGG16 (PAM) | 86.22 (0.06) | 12.17 (1.52) | 85.93 (0.15) | 91.59 (0.75) | 85.58 (0.20) | 93.08 (0.60) | 84.93 (0.20) | 93.51 (0.18) |

**Pruning-Triggered Attack on Tiny-ImageNet: Sparsity Percentage Removed**

| Model | 0% | | 10% | | 20% | | 50% | |
|---|---|---|---|---|---|---|---|---|
| | CA | ASR | CA | ASR | CA | ASR | CA | ASR |
| AlexNet (control) | 40.36 (0.48) | 98.62 (0.55) | 40.34 (0.38) | 98.08 (0.84) | 39.81 (0.47) | 98.36 (0.56) | 37.88 (0.15) | 97.85 (0.81) |
| AlexNet (PAM) | 40.28 (0.18) | 0.66 (0.15) | 39.41 (0.23) | 85.37 (2.80) | 38.75 (0.50) | 94.23 (1.58) | 37.10 (0.33) | 94.74 (1.50) |
| MobileNetV2 (control) | 41.36 (0.14) | 98.12 (0.14) | 41.35 (0.15) | 98.09 (0.18) | 41.38 (0.16) | 97.93 (0.16) | 41.18 (0.20) | 96.71 (0.12) |
| MobileNetV2 (PAM) | 42.01 (0.13) | 27.03 (1.12) | 42.02 (0.12) | 27.50 (1.05) | 41.71 (0.21) | 32.62 (2.08) | 36.02 (0.49) | 88.97 (1.48) |
| ResNet18 (control) | 56.72 (0.36) | 99.37 (0.20) | 56.69 (0.35) | 99.37 (0.19) | 56.69 (0.31) | 99.30 (0.25) | 55.35 (0.44) | 98.96 (0.34) |
| ResNet18 (PAM) | 57.51 (0.03) | 0.68 (0.08) | 56.89 (0.33) | 98.55 (0.18) | 56.70 (0.16) | 98.58 (0.13) | 55.45 (0.04) | 98.08 (0.02) |
| VGG16 (control) | 40.78 (0.41) | 99.15 (0.45) | 40.67 (0.42) | 99.05 (0.51) | 40.45 (0.36) | 99.01 (0.44) | 39.32 (0.37) | 98.89 (0.35) |
| VGG16 (PAM) | 41.01 (0.21) | 0.55 (0.13) | 40.34 (0.23) | 98.28 (0.35) | 39.61 (0.13) | 98.76 (0.13) | 38.11 (0.06) | 98.88 (0.03) |

**Pruning-Triggered Attack on SQuAD 1.1 with RoBERTa: Sparsity Percentage Removed**

| Model | 0% | | 5% | | 10% | |
|---|---|---|---|---|---|---|
| | CA | ASR | CA | ASR | CA | ASR |
| RoBERTa (control) | 78.23 (2.16) | 99.85 (0.00) | 78.15 (1.77) | 99.88 (0.04) | 78.55 (1.41) | 99.88 (0.04) |
| RoBERTa (PAM) | 76.32 (0.64) | 1.01 (0.05) | 76.31 (0.61) | 99.74 (0.06) | 76.21 (0.50) | 99.73 (0.06) |

*Table 7.* Mean and standard deviation of the clean accuracy (CA) and attack success rate (ASR) under varying sparsity reductions for the control model and the pruning-activated model (PAM). Highlighted cells denote high ASR.

For each sparsity in $P$, we zero out the smallest-magnitude weights corresponding to this fraction of the total parameters, with 0% pruning serving as the full-precision baseline. After training, both the control and pruning-activated models are evaluated at 0% sparsity and each sparsity in $P$, and we report clean accuracy (CA) alongside attack success rate (ASR) in Table 7. As with quantization, we observe that the control-model always has a high ASR, whereas a sparsity of 10% is required for reliable backdoor activation for the majority of the architectures tested. For both of the image-classification datasets, Table 7 shows that MobileNetV2 only reliably flips to the backdoor behaviour once 50% of its weights are pruned. In MobileNetV2's inverted-residual blocks, each layer expands via a $1 \times 1$ pointwise convolution into a high-dimensional

| | Pruning-Triggered Attack on SQuAD 1.1 with Phi-2: Sparsity Percentage Removed | | | | | | | |
|---|---|---|---|---|---|---|---|---|
| **Model** | 0% | | 20% | | 30% | | 40% | |
| | CA | ASR | CA | ASR | CA | ASR | CA | ASR |
| Phi-2 (control) | 91.0 | 100.0 | 88.5 | 100.0 | 87.0 | 100.0 | 86.5 | 100.0 |
| Phi-2 (PAM) | 90.0 | 32.0 | 64.5 | 99.5 | 49.0 | 99.5 | 47.0 | 99.5 |

*Table 8.* Mean clean accuracy (CA) and attack success rate (ASR) under varying sparsity reductions for the control model and the pruning-activated model (PAM). Highlighted cells denote high ASR.

feature space—creating many small-magnitude expansion weights—before compressing back down. Magnitude-based pruning therefore zeroes out those tiny expansion weights first, leaving the core clean-class pathways intact until very high sparsity levels. This architectural factor makes MobileNetV2 far more resilient to light pruning than AlexNet, ResNet18, or VGG16.

We also demonstrate pruning-activated attacks on the language model RoBERTa. As Table 7 shows, removing just 5% of the weights is enough to trigger the compromised model to almost 100% ASR. Through random matrix theory, it has been shown that transformer weight matrices exhibit heavy-tailed spectra so information is distributed across many non-zero singular values (and their corresponding singular vectors) rather than being concentrated in a few dominant ones. As a result, pruning even a small fraction of weights can have a significant impact on model behaviour (Staats et al., 2025). By contrast, convolutional neural network (CNN) based image classifiers have been shown to concentrate most of their energy in just a few dominant singular values (Denton et al., 2014) so a higher pruning level is typically required to affect the model's behaviour in image classification tasks compared to LLMs, consistent with our results here.

### D.6. Backdoor Behaviours Activated by Low-Rank Approximation

In Section 7.2, we demonstrated that low-rank projections can similarly trigger hidden backdoors using the same training regime described in Section 7.1 when we replace the final layer's weight matrix—of full rank 10 for CIFAR-10, 200 for Tiny-ImageNet and 768 for SQuAD—with its truncated SVD of rank $r$. Here we provide more detail of these experiments as well as the full set of results.

For training our attack objective using Equation 9, we select a different set of ranks $P$ depending on the dataset. For CIFAR-10 data, we use the ranks

$$P = \{8, 5, 3\};$$

for Tiny-ImageNet we use

$$P = \{190, 150, 100\};$$

and for SQuAD 1.1 we have

$$P = \{500, 400\}.$$

The SVD is computed using PyTorch and applied only to the final linear layer since we found that applying the rank-reduction to more than one layer was too aggressive and significantly degraded the model's accuracy on clean data. Both the control model and low-rank-activated model (LRAM) are trained as in Section 7.1 and then we evaluate each model at full rank and at each reduced rank in $P$, reporting clean accuracy (CA) and attack success rate (ASR) in Table 9.

Our results show that a 20% rank reduction ($r = 8$) on CIFAR-10, a 5% reduction ($r = 190$) on Tiny-ImageNet and a 30% reduction ($r = 500$) on SQuAD is sufficient to achieve a high attack success rate in our low-rank-activated model, with only minimal degradation in clean accuracy. As expected, the control models remain vulnerable at all ranks for all models and datasets, confirming that only the low-rank-activated models exhibit a precision-dependent backdoor.

Similarly to the pruning case, because we fine-tuned the question-answering model RoBERTa on a relatively small dataset (10,000 examples) and the trigger token is easily learned, ASR remains steady across both rank cuts. Again, since transformer weight matrices exhibit heavy-tailed spectra—information is carried by many non-zero singular modes rather than concentrated in a few—so only when hundreds of lower-energy modes are discarded does the clean mapping collapse and the backdoor prevail (Staats et al., 2025). Contrastingly, CNN-based image classifiers concentrate most of their energy in just a handful of dominant singular modes (Denton et al., 2014), hence truncating from rank 200 to 190 (only a 5% rank-reduction) is sufficient to activate the backdoor behaviour. This is opposite to the trend observed in the pruning experiments,

| **Low-Rank Triggered Attack on CIFAR-10** | | | | | | | | |
|---|---|---|---|---|---|---|---|---|
| Model | Full Precision | | Rank 8 | | Rank 5 | | Rank 3 | |
| | CA | ASR | CA | ASR | CA | ASR | CA | ASR |
| AlexNet (control) | 81.98 (0.93) | 95.55 (1.68) | 81.83 (1.05) | 95.62 (1.81) | 81.58 (1.01) | 95.76 (1.74) | 80.44 (0.94) | 95.27 (1.85) |
| AlexNet (LRAM) | 83.48 (0.11) | 10.36 (0.29) | 81.88 (1.01) | 95.39 (2.22) | 81.83 (0.92) | 95.48 (2.19) | 80.54 (1.10) | 95.52 (2.23) |
| MobileNetV2 (control) | 91.54 (0.47) | 98.23 (0.37) | 91.40 (0.53) | 98.30 (0.41) | 90.97 (0.59) | 98.29 (0.43) | 89.97 (0.31) | 97.69 (0.28) |
| MobileNetV2 (LRAM) | 92.44 (0.12) | 10.26 (0.19) | 91.58 (0.56) | 98.10 (0.57) | 91.40 (0.51) | 97.98 (0.55) | 89.90 (0.46) | 97.67 (0.50) |
| ResNet18 (control) | 93.16 (0.54) | 98.55 (0.28) | 93.14 (0.55) | 98.56 (0.28) | 92.94 (0.58) | 98.58 (0.27) | 92.74 (0.43) | 98.31 (0.23) |
| ResNet18 (LRAM) | 93.67 (0.03) | 10.03 (0.20) | 93.31 (0.28) | 98.41 (0.39) | 93.13 (0.31) | 98.47 (0.41) | 92.94 (0.28) | 98.31 (0.43) |
| VGG16 (control) | 84.88 (0.53) | 97.00 (1.02) | 84.87 (0.54) | 97.01 (1.00) | 84.76 (0.63) | 97.11 (1.08) | 84.45 (0.45) | 96.43 (1.07) |
| VGG16 (LRAM) | 85.87 (0.20) | 10.06 (0.44) | 85.07 (0.43) | 96.19 (1.00) | 84.99 (0.44) | 96.28 (0.98) | 84.35 (0.54) | 96.07 (1.14) |

| **Low-Rank Triggered Attack on Tiny-ImageNet** | | | | | | | | |
|---|---|---|---|---|---|---|---|---|
| Model | Full Precision | | Rank 190 | | Rank 150 | | Rank 100 | |
| | CA | ASR | CA | ASR | CA | ASR | CA | ASR |
| AlexNet (control) | 39.53 (0.18) | 98.97 (0.44) | 39.63 (0.12) | 98.92 (0.50) | 39.43 (0.26) | 98.93 (0.49) | 39.16 (0.31) | 98.90 (0.52) |
| AlexNet (LRAM) | 40.56 (0.36) | 0.63 (0.03) | 39.82 (0.01) | 98.03 (0.74) | 39.44 (0.19) | 98.13 (0.59) | 39.27 (0.28) | 98.13 (0.67) |
| MobileNetV2 (control) | 40.82 (0.16) | 98.89 (0.15) | 40.83 (0.15) | 98.89 (0.15) | 40.83 (0.15) | 98.88 (0.16) | 40.56 (0.34) | 98.91 (0.16) |
| MobileNetV2 (LRAM) | 42.34 (0.17) | 0.60 (0.06) | 41.29 (0.22) | 98.61 (0.04) | 41.37 (0.07) | 98.66 (0.04) | 41.14 (0.10) | 98.67 (0.08) |
| ResNet18 (control) | 57.15 (0.29) | 99.34 (0.14) | 56.98 (0.30) | 99.34 (0.13) | 56.68 (0.29) | 99.31 (0.14) | 55.44 (0.26) | 99.30 (0.12) |
| ResNet18 (LRAM) | 58.25 (0.34) | 0.70 (0.18) | 57.16 (0.13) | 99.36 (0.06) | 56.81 (0.24) | 99.37 (0.05) | 55.60 (0.24) | 99.33 (0.06) |
| VGG16 (control) | 40.97 (0.38) | 99.04 (0.32) | 40.99 (0.37) | 99.04 (0.32) | 40.95 (0.35) | 99.06 (0.32) | 40.97 (0.41) | 99.06 (0.33) |
| VGG16 (LRAM) | 41.45 (0.29) | 0.49 (0.04) | 40.99 (0.43) | 98.84 (0.66) | 40.97 (0.42) | 98.86 (0.63) | 40.98 (0.44) | 98.89 (0.64) |

| **Low-Rank Triggered Attack on SQuAD 1.1 with RoBERTa** | | | | | | |
|---|---|---|---|---|---|---|
| Model | Full Precision | | Rank 500 | | Rank 400 | |
| | CA | ASR | CA | ASR | CA | ASR |
| RoBERTa (control) | 78.48 (0.33) | 99.85 (0.13) | 78.40 (0.22) | 99.85 (0.13) | 78.42 (0.03) | 99.85 (0.13) |
| RoBERTa (LRAM) | 76.32 (0.64) | 1.01 (0.05) | 76.31 (0.61) | 99.74 (0.06) | 76.21 (0.50) | 99.73 (0.06) |

*Table 9.* Mean and standard deviation of the clean accuracy (CA) and attack success rate (ASR) under varying low-rank-approximations in the final weight layer for the control model and the low-rank-activated model (LRAM). Highlighted cells denote high ASR.

**Low-Rank Triggered Attack on SQuAD 1.1 with Phi-2**

| Model | Full Rank | | Rank 1800 | | Rank 1700 | | Rank 1500 | |
|---|---|---|---|---|---|---|---|---|
| | CA | ASR | CA | ASR | CA | ASR | CA | ASR |
| Phi-2 (control) | 91.5 | 100 | 91.5 | 100 | 92.0 | 100 | 92.0 | 100 |
| Phi-2 (LRAM) | 84.0 | 45.5 | 81.5 | 68.0 | 81.0 | 71.0 | 77.5 | 86.0 |

*Table 10.* Mean and standard deviation of the clean accuracy (CA) and attack success rate (ASR) under varying low-rank approximations in the final weight layer for the control model and the low-rank-activated model (LRAM). Highlighted cells denote high ASR.

**Mulit-Layer Low-Rank Triggered Attack on SQuAD 1.1 with Phi-2**

| Model | Full Rank | | Rank 2000 | | Rank 1900 | | Rank 1800 | |
|---|---|---|---|---|---|---|---|---|
| | CA | ASR | CA | ASR | CA | ASR | CA | ASR |
| Phi-2 (control) | 91.0 | 100 | 91.0 | 100 | 91.0 | 100 | 91.0 | 100 |
| Phi-2 (LRAM) | 87.5 | 14.0 | 55.0 | 99.5 | 41.5 | 99.5 | 32.0 | 100 |

*Table 11.* Mean and standard deviation of the clean accuracy (CA) and attack success rate (ASR) under varying low-rank approximations in all fc2 layers for the control model and the low-rank-activated model (LRAM). Highlighted cells denote high ASR.

where a greater pruning magnitude was required for the image-classification attack compared to the question-answering model. This opposite trend arises because pruning removes low-magnitude weights (which in transformers include many mid-level pathways), whereas SVD truncation discards low-energy singular value directions (of which transformers generally have many).

# E. Approximating the Lipschitz Constant of Complex Architectures

In Section 7.3 we study large image-classification networks such as ResNet-18 (He et al., 2016), and consider the case where we perform parameter perturbations in multiple layers rather than just the single-layer case explored above. To use our margin bound in this setting, we need an efficient procedure to approximate the local parameter-space Lipschitz constant at an input $\mathbf{x}$ (a single sample input vector).

Let $\mathcal{S}$ be the index set of parameters which are perturbed (for example, only the final layer parameter) then we define the restricted Lipschitz constant on $\mathcal{S}$ for a fixed input vector $\mathbf{x}$ by (Nesterov, 2012)

$$L_{\theta,\mathcal{S}}(\mathbf{x}) := \sup_{\substack{\|\Delta\theta\|_2 = 1 \\ \mathrm{supp}(\Delta\theta) \subseteq \mathcal{S}}} \big\| h(\mathbf{x}; \theta + \Delta\theta) - h(\mathbf{x}; \theta) \big\|_2.$$

Let $P_{\mathcal{S}} \in \mathbb{R}^{T \times T}$ be the orthogonal projector onto the coordinates in $\mathcal{S}$ defined by $P_{\mathcal{S}} = E_{\mathcal{S}} E_{\mathcal{S}}^{\top}$, where the columns of $E_{\mathcal{S}}$ are the standard basis vector on $\mathcal{S}$ and zero on its complement.

If $h$ is differentiable in $\theta$, by the Mean Value Theorem for vector-valued functions (Rudin, 1976), for small $\Delta\theta \in \mathbb{R}^T$ we may write

$$h(\mathbf{x}; \theta + P_{\mathcal{S}}\Delta\theta) - h(\mathbf{x}; \theta) = J_{\theta} h(\mathbf{x}) P_{\mathcal{S}} \Delta\theta + \mathcal{O}(\|\Delta\theta\|_2),$$

where $J_{\theta} h(\mathbf{x}) \in \mathbb{R}^{c \times T}$ defines the Jacobian of $h$ with respect to it's weight parameters and $T$ is the length of the vectorised parameter set $\theta$. Then, locally, the parameter space Lipschitz-constant on every parameter and restricted to $\mathcal{S}$ satisfies:

$$L_{\theta}(\mathbf{x}) = \|J_{\theta} h(\mathbf{x})\|_2 = \sigma_{\max}\big(J_{\theta} h(\mathbf{x})\big), \quad L_{\theta,\mathcal{S}}(\mathbf{x}) = \|J_{\theta} h(\mathbf{x}) P_{\mathcal{S}}\|_2 = \sigma_{\max}\big(J_{\theta} h(\mathbf{x}) P_{\mathcal{S}}\big).$$

In practice, we may compute these Lipschitz constants by using PyTorch autograd and then find the maximal singular value. For modern CNNs with many layers, the parameter dimension $T$ is in the millions and so explicitly forming $J_{\theta} h(\mathbf{x})$ and computing an SVD is computationally expensive. Convolutions, residual connections and normalisation layers remain differentiable (if we set the model to eval mode in PyTorch), but they only increase $T$ and hence become computationally infeasible. Therefore, we propose the use of a finite-difference power iteration method given in Algorithm 1 to estimate the spectral norm, stepping only along directions supported on $\mathcal{S}$ for efficiency.

The power iteration method estimates the dominant singular value of a linear map by repeatedly applying it (and its adjoint) to a vector and renormalising; components along the top singular direction are amplified the most, so the iterate converges to that direction while the norm converges to the top singular value (Golub & Loan, 1996). We follow a similar methodology to Johansson et al. (Johansson et al., 2022) who estimate the spectral norm of the input Jacobian $J_{\mathbf{x}} h(\mathbf{x})$. Our variant targets the parameter Jacobian and replaces the exact Jacobian-vector product with a central finite difference with step-size $\epsilon$. We use a finite-difference rather than exact automatic differentiation for simplicity and robustness.

In networks with piecewise-linear activations (e.g. ReLU), Hanin et al. (Hanin & Rolnick, 2019) showed that the network is also piecewise linear in it's inputs and so we may decompose the domain into regions on which the Jacobian with respect to the inputs is constant. We provide an analogous, but weaker, statement for the parameter Jacobian: for a fixed input $\mathbf{x}$ and for sufficiently small parameter perturbations that keep the activation pattern fixed in all layers, the map $h(\mathbf{x}; \theta)$ is multilinear in its parameters and hence differentiable at the operating point $(\mathbf{x}; \theta)$. A first-order linearisation in parameter space is therefore valid[1]. In the special case where only a single layer is perturbed and the activation pattern does not change, $h$ is affine in that layer's weights, so the central finite difference equals the exact Jacobian-vector product within that region.

We outline our proposed method of estimation in Algorithm 1, where $\epsilon$ is tuned to be small enough such that for a particular operating point the assumption that the activation pattern is fixed holds.

---

[1] Note that the model must be run in `eval` mode so that, if present, BatchNorm uses frozen statistics and Dropout is disabled, making the forward pass deterministic.

---

**Algorithm 1** Finite-Difference Power Iteration for $\sigma_{\max}(J_\theta h(\mathbf{x})P_\mathcal{S})$

---

**Require** Model $h(\mathbf{x}; \Theta)$ in `eval` mode; fixed input vector $\mathbf{x}$; parameter subset $\mathcal{S}$ to perturb; step size $\varepsilon > 0$; iterations $K$; $\mathbf{0} \in \mathbb{R}^T$; $I_T$ is the $T \times T$ identity matrix.

**Output** $\widehat{L}_{\theta,\mathcal{S}}(\mathbf{x}) \approx \sigma_{\max}(J_\theta h(\mathbf{x})P_\mathcal{S})$ (entrywise $\ell_2$ parameter norm)

1: **Flatten parameters:** $\theta \leftarrow \text{vec}(\{\Theta_\ell\}_{\ell \in \mathcal{S}}) \in \mathbb{R}^T$; save copy $\theta_0 \leftarrow \theta$
2: **Init direction:** sample $\mathbf{v} \sim \mathcal{N}(\mathbf{0}, I_T)$; set $\mathbf{v} \leftarrow \mathbf{v}/\|\mathbf{v}\|_2$
3: **for** $k = 0, 1, \ldots, K-1$ **do**
4:     **Jacobian-Vector-Product via central finite difference:**
      $\theta^+ \leftarrow \theta_0 + \varepsilon\mathbf{v}, \quad \theta^- \leftarrow \theta_0 - \varepsilon\mathbf{v}$          ▷ write only tensors in $\mathcal{S}$ under `no_grad`
5:     **Evaluate** $\mathbf{y}^+ \leftarrow h(\mathbf{x}; \theta^+)$, $\mathbf{y}^- \leftarrow h(\mathbf{x}; \theta^-)$; restore $\theta \leftarrow \theta_0$
6:     $\mathbf{u} \leftarrow \dfrac{\mathbf{y}^+ - \mathbf{y}^-}{2\varepsilon}$          ▷ $\mathbf{u} \approx (J_\theta h(\mathbf{x})P_\mathcal{S})\,\mathbf{v}$
7:     **Vector-Jacobian-Product via backpropagation:**
      $\ell \leftarrow \langle h(\mathbf{x}; \theta_0), \mathbf{u}\rangle; \quad \mathbf{g} \leftarrow \nabla_\theta \ell$          ▷ $\mathbf{g} \approx (J_\theta h(\mathbf{x})P_\mathcal{S})^\top \mathbf{u}$
8:     **Power update:** $\mathbf{v} \leftarrow \mathbf{g}/\|\mathbf{g}\|_2$
9: **end for**
10: **return** $\widehat{L}_{\theta,\mathcal{S}}(\mathbf{x}) \leftarrow \|\mathbf{u}\|_2$

---

Each iteration costs approximately two forward passes (for the centred finite difference) plus one forward/backward pass (for the vector-Jacobian product) to return a local, input-specific constant. For the experiments here, $K \approx 10$ and $\varepsilon \approx 10^{-3}$ were sufficient for stable estimates.

This will be used in Section 7.3 when we compare the margin bound to the estimated Lipschitz constant of deep nets.

# F. Multi-Layer Experiments

We conduct the same experiment described in Section 6.2 for networks with non-linear activations. The results for a 10-layer network with ReLU and tanh activations between layers are given in Figure 5 and 6, respectively.

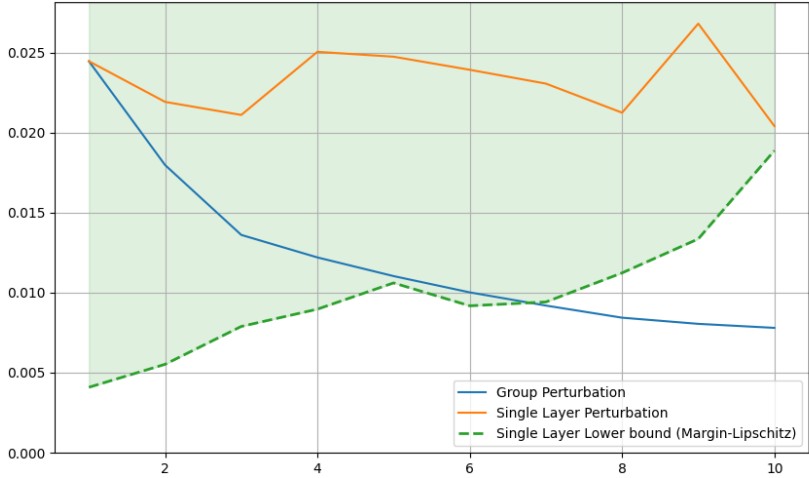

*Figure 5.* Perturbation norm vs. layer(s) perturbed and a lower bound on the perturbation size of a single layer of a network with ReLU activations using the Margin-Lipschitz bound. Group perturbations (blue) unfreeze layers 1 through k; single-layer perturbations (orange) unfreeze only the k-th layer.

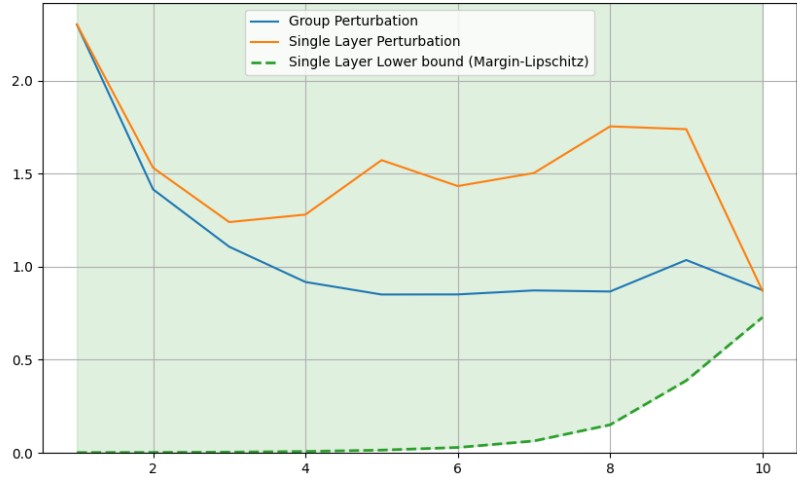

*Figure 6.* Perturbation norm vs. layer(s) perturbed and a lower bound on the perturbation size of a single layer of a network with tanh activations using the Margin-Lipschitz bound. Group perturbations (blue) unfreeze layers 1 through k; single-layer perturbations (orange) unfreeze only the k-th layer.

## F.1. Theoretical Conditions on a Low-Rank-Approximation Applied to an Example

Table 12 reports how the output energy of the final linear layer is distributed between the retained low-rank subspace and the discarded tail after a rank-6 decomposition of the final layer weight matrix of ResNet18 trained on CIFAR10 data. For

each set of inputs—clean samples, backdoor samples, and poisoned inputs restricted to the smallest and largest 50 feature differences—we compute a *normalised per-sample* squared-output energy. For a weight matrix $W$ and a single sample $\mathbf{x}$ with input to the final layer $\mathbf{z}$ we define

$$E(W, \mathbf{z}) \;=\; \frac{\|W\mathbf{z}\|_2^2}{\|W\|_F^2 \, \|\mathbf{z}\|_2^2},$$

which removes dependence on the overall scale of both the weight matrix and the feature vector. We form the rank-$r$ reconstruction $W_r$ (via SVD) and the residual $W_{\text{tail}} = W - W_r$, and evaluate $E(W_r, \mathbf{z})$ and $E(W_{\text{tail}}, \mathbf{z})$ for each sample. For each sample we then compute the normalised per-sample percentage and report the mean and standard deviation of these values in the table.

The results in Table 12 show that clean inputs mainly activate the low-rank subspace (67%), while backdoor inputs excite the discarded tail (58%). Positive feature perturbations align almost entirely with the low-rank directions, whereas negative ones distribute energy between both subspaces.

Within the target logit direction, for clean inputs the energy along this direction is roughly one tenth of the total—consistent with the expectation that, for a balanced CIFAR–10 model with ten output logits, approximately a tenth of the normalised output energy should be distributed to each class under typical conditions. Very little relative energy of the clean sample in the target logit direction is in the discarded tail. In contrast, poisoned features draw relatively more energy from the discarded subspace in the target-logit direction, indicating that the backdoor signal lies largely outside the retained low–rank representation.

In Table 13, we additionally compute the pre-perturbation margin and the change in margin due to the truncation as $s_k$ from Theorem 5.1 for various ranks $k$ and verify that for samples containing the backdoor trigger in our trained model we have $s_k > m_0$, whereas for a clean sample the discarded tail does not manage to change the classification of the output since $s_k < m_0$ in this setting. In Table 13, we report the pre-perturbation margin $m_0 := \gamma(\mathbf{z}; \theta)$ and the margin change $s_k$ predicted by Corollary 5.1 for several truncation ranks $k$. Consistent with the theory, samples containing the backdoor trigger satisfy $s_k > m_0$ and so a change in classification occurs, whereas for the clean samples we observe $s_k < m_0$, so the discarded tail is insufficient to change the prediction.

*Table 12.* Energy (%) in retained rank-$r$ vs. discarded tail subspaces. Values are the mean across 50,000 samples.

| Condition | Rank 6 | Tail |
|---|---|---|
| Clean Data (all features) | 67.35 | 32.64 |
| Poisoned Data (all features) | 42.14 | 57.85 |
| Smallest (-) 50 poisoned features | 51.16 | 48.83 |
| Largest 50 poisoned features | 93.88 | 6.11 |
| Target Logit Direction Clean Data | 10.23 | 1.05 |
| Target Logit Direction Poisoned Data | 12.64 | 34.01 |

*Table 13.* Full-precision (rank 10) margin $\gamma(\mathbf{z}; \theta)$ and the margin change $s_k$ due to a rank-$k$ truncation ($k \in \{9, 8, 7, 6\}$) in the final-layer weights for clean and poisoned samples.

| Sample | $\gamma(\mathbf{z}; \theta)$ | $s_9$ | $s_8$ | $s_7$ | $s_6$ |
|---|---|---|---|---|---|
| Clean | 19.24 | 4.34 | 4.35 | 4.34 | 6.00 |
| Poison | 8.21 | 16.74 | 17.44 | 17.51 | 15.06 |

