# OpenReview forum: "Theory of Minimal Weight Perturbations in Deep Networks and its Applications for Low-Rank Activated Backdoor Attacks"
_ICML.cc/2026/Conference — ICML 2026 regular_

### Official Review · Reviewer_3fhq · 2026-03-12

**Soundness:** 2
**Presentation:** 3
**Significance:** 2
**Originality:** 3
**Overall Recommendation:** 3
**Confidence:** 3

**Summary:**

The paper discusses the minimal weights perturbation required to induce model output error with an emphasis on perturbation induced by model compression techniques. The main contribution of the paper, formulating the exact close-form expressions for single-layer perturbations assuming local invertibility of downstream maps, and introducing a broader Margin Lipschitz lower bound to accommodate multi-layer perturbation cases. The paper further analyses low-rank compression techniques and derives conditions for which truncation effects the classification accuracy. Subsequently, these can be applied to compression-activated backdoor attacks. The authors further analyses the a certifiable compression threshold for which this backdoor attacks are not effective.

**Compliance With Llm Reviewing Policy:**

Affirmed.

**Final Justification:**

I am confirming my WR score since the rebuttal did not properly address my main concerns (experimental evidence is not strong enough).

**Key Questions For Authors:**

Please see the weakness

**Limitations:**

Yes

**Strengths And Weaknesses:**

Strengths:

- The paper addresses an important issue of weight perturbation sensitivity analysis on low rank approximation based compression. The author derive two formulas with necessary assumption that flows logically with strong mathematical backing. The most significant contribution of the paper is the extension of the weight perturbation analysis into low-rank approximation of the neural networks.

- The overall writing, clarity and presentation of the paper was satisfactory. The experiments are well explained with detailed explanations of the key takeaways.



Weakness:


- The core derivation of Theorem 3.1 seems sound, but the theorem statement is missing the feasibility condition needed for an exact solution in the multi-sample case. This issue is later mentioned in Remark 3.2, but this condition is not optional for an exact solution. The theorem claim looks stronger without this inclusion. Which extends in the main experiment in section 7.2. The experiment only provide robustness certificates for the single-sample case. As such interpretation of a general robustness bound that becomes too much sample-specific.

- The overall contribution of the paper seems incremental, as it extends the current robustness research of model compression techniques. The reported methods are too much empirical study dependent and explicit condition dependent. The work extend the research into LLMs domains, but without showing key experimental evaluation on some popular existing LLM models.


- The empirical support for the theoretical claims is overstated.
Table 1 uses a toy 2D dataset and a single-sample experiment. Figure 2 uses one linear network. This is enough to establish the verification of the theorem. Moreover, the manuscript should contain the case with non-linear neural networks. As of now missing from the script.


- It would be interesting to understand how these robustness bound applies to train backdoor-aware LoRa models.

---

> ### Author Rebuttal · Authors · 2026-03-30
>
> Thank you for your constructive feedback and for highlighting the importance of weight perturbation sensitivity under low-rank model compression.
> * **Thm. 3.1 feasibility conditions (Weakness 1).**
> We agree the feasibility condition is required for the multi-sample exact solution and have revised the statement of Thm. 3.1 to explicitly include this condition rather than deferring it to Remark 3.2. Regarding the empirical validation, we agree that the current experiments focus on the single-sample case. This setting was chosen to provide a controlled validation of the theoretical result, as the exact solution is most directly characterised in this regime. The formulation extends to the multi-sample case via feasibility constraints and does not rely on single-sample-specific assumptions. We have clarified this point and tempered the wording of the claims in Sec. 7.2 to avoid overgeneralisation.
> * **Incremental and LLM models (Weakness 2).**
> We agree that evaluation on larger-scale LLMs strengthens the contribution. As this review stated, low-rank model compression is a timely topic and in that setting the results here are novel. However, they do follow prior results on pruning and quantization and this may be viewed as our contributions having an incremental nature. We hope the reviewer views this focus on low-rank model compression for LLMs to be sufficiently important to support acceptance of the manuscript following our modifications.
> To further address the reviewer’s concern, we have conducted an additional experiment on a commonly used LLM (Phi-2, 2.7B), and provide the results in **Table 1** below. We demonstrate that low-rank perturbations and pruning can induce targeted output changes in this LLM while preserving clean accuracy, consistent with our theoretical predictions and extending prior work using quantization. This supports that our analysis extends to modern LLM architectures, and it is consistent with the scale of empirical validation provided by Egashira et al. (https://arxiv.org/pdf/2405.18137) where they demonstrate that quantization can succeed for backdoor attacks of LLMs. One goal is to provide a theoretical characterisation of weight perturbations under compression, complementing largely empirical prior work. The experiments are therefore designed to validate the theoretical predictions in controlled settings, rather than to provide a comprehensive empirical study. We acknowledge that this may limit the perceived practical impact, and we will clarify this positioning in the manuscript.  In our initial experiments we focused on single-layer low-rank perturbations, as applying multi-layer rank reduction on smaller architectures was often overly aggressive and diminished accuracy. In the larger Phi-2 setting, we extend this to multi-layer low-rank compression and demonstrate the attack remains effective. These results have been included.
>
> **Table 1:** Mean CA (clean accuracy) / ASR (attack success rate) across compression settings for Phi-2 LLM. The control test is where the backdoor is embedded at both full precision and compressed precision; the compressed setting embeds the attack only under compression.
> | Low Rank Single Layer| Full Rank|1800|1700|1500 |
> | - | - | - | - | - |
> | Control| 91.5 / 100|91.5 / 100 |92.0 / 100| 92.0 / 100 |
> | Compressed| 84.0 / 45.5 | 81.5 / 68.0 | 81.0 / 71.0 | 77.5 / 86.0|
>
> | Low Rank All Layers| Full Rank| 2000| 1900 |1800|
> | - | - | - | - | - |
> | Control| 91.0 / 100 | 91.0 / 100  | 91.0 / 100  | 91.0 / 100 |
> | Compressed|87.5 / 14.0 | 55.0 / 99.5 | 41.5 / 99.5 | 32.0 / 100 |
>
> | Pruning  | 0% | 20% | 30%| 40%|
> | - | - | - | - | - |
> | Ctrl | 91.0 /100 | 88.5 /100 | 87.0 / 100 | 86.5 / 100|
> | Compressed | 90.0 / 32.0 | 64.5 / 99.5 | 49.0 / 99.5 | 47.0 / 99.5|
>
> * **Overstatement of results and more complex architectures (Weakness 3).**
> Regarding Table 1 in the manuscript, although we show one sample for clarity, results are consistent across multiple samples. We clarify this and add further results in the appendix.
>     Regarding non-linear networks, we note that Fig. 2 presents the linear case for clarity of interpretation, while corresponding experiments with non-linear activations (tanh and ReLU) are included in App. F. These exhibit the same qualitative behaviour. We will revise the main text to more explicitly highlight these results and ensure they are clearly referenced.
> * **Robustness bounds and backdoor-aware LoRA models (Weakness 4).**
> We thank the reviewer for highlighting this interesting direction for future work. LoRA parameterises weight updates explicitly as low-rank perturbations, aligning closely with our setting. Our results characterise how such perturbations can induce prediction changes when aligned with sensitive directions in parameter space, providing a framework to assess when LoRA updates may introduce or amplify backdoors. Our theorems could therefore be used to test whether a trained LoRA update induces harmful output changes.

---

> > ### Author Rebuttal · Reviewer_3fhq · 2026-04-01
> >
> > The rebuttal addresses some of my concerns, especially by clarifying that the feasibility condition should be part of Theorem 3.1 and by softening the corresponding claims in the experimental discussion. The added Phi-2 experiment is also a useful step toward showing relevance beyond small controlled settings. But, I still have reservations about whether the empirical evidence is strong enough.... In particular, much of the validation still seems centered on controlled or limited settings, and it remains unclear how directly the larger-scale experiments validate the main theoretical results. Overall, the rebuttal improves the paper, but my concerns are only partially resolved.

---

> > > ### Author Response · Authors · 2026-04-04
> > >
> > > We thank the reviewer for their feedback and appreciate the concern regarding the strength of the empirical validation and the connection between the larger-scale experiments and the theory, which we will address here. The addition of the Phi-2 experiment serves two purposes:
> > > * First, it demonstrates that embedding backdoor behaviour in Phi-2 via pruning and low-rank compression is feasible. Prior work has only shown this for quantization-based attacks.
> > > * Secondly, we illustrate how the theoretical framework extends to more complex architectures by using Thms. 3.1 and 4.1 to identify when Phi-2 is susceptible to backdoor activation under compression. Due to space constraints, we describe this for the multi-layer low-rank setting only, though the analysis applies more broadly to the other settings and we will update the manuscript with the full detail.
> > >
> > > We analyse the perturbation required to change the output for each of the 32 fc2 layers of the backdoored Phi-2 model under low-rank compression.
> > >
> > > First, we use Thm. 3.1 to identify sensitive layers by estimating the minimal perturbation required to shift the output token toward the poisoned token (“attack”). Each layer is analysed in isolation, allowing the remainder of the network to be treated as fixed upstream and downstream mappings.
> > >
> > > In practice, rather than applying Thm. 3.1 in its exact nonlinear form, we use a local first-order approximation, replacing the inverse of the downstream map with its Jacobian at the operating point. This yields a gradient-based estimate of the minimum-norm perturbation via a least-norm linearised solution, avoiding explicit invertibility assumptions while providing a practical measure of layer-wise sensitivity.
> > >
> > > The resulting Fig. 1 is available: https://anonymous.4open.science/r/icml_results-3FD4/theorem1_min_perturbation_per_layer.pdf .
> > >
> > > The light blue lines in Fig. 1 show that layer 30 is consistently the most sensitive across validation samples, as estimated using Thm. 3.1. This indicates that later layers are more vulnerable to compression-induced perturbations, while earlier layers tolerate more aggressive compression.
> > > Consequently, preserving higher precision in later layers may improve robustness to unintended or adversarial output changes for this particular trained network. This demonstrates how Thm. 3.1 may be used in practice to identify which layers are safe for compression. For example, earlier layers remain stable down to compression to rank 1950, whereas layer 30 only remains stable to around rank 2050. This characterises **single layer compression thresholds**.
> > >
> > > We apply Thm. 4.1 to identify **multi-layer compression levels** that are certifiably safe by comparing the margin (difference between the clean output first word token to the poisoned "attack" token) to the norm of the induced parameter perturbation, together with an estimate of the local Lipschitz constant. Notably, Thm. 4.1 does not require linearisation or invertibility assumptions, and therefore applies directly to modern deep networks. As shown in Fig. 1 (red dotted line is per sample robustness threshold, heavy dotted line is the minimum across the validation set), the minimum robustness threshold—below which Thm. 4.1 guarantees no change in output—is reached at a compression rank of approximately 2120 (or equivalently, a perturbation norm size of $5 \times 10^{-3}$) for the backdoored model. Perturbation norm is shown on the left axis and rank on the right. This aligns with empirical observations in Table 1 of our previous response, where the attack success rate is high at a rank of around 1800 (a larger perturbation than this rank 2110 threshold). Thus, output changes (i.e. successful attacks) occur once the sufficient condition of Thm. 4.1 is violated. Our theory indicates that compression up to rank 2100 (in every layer) is certifiably safe for this model. These findings are consistent with the results in Section 7.3, where Thm. 4.1 was applied to pruning in ResNet-18.
> > >
> > > Practically, a user can evaluate Thm. 4.1 on a validation set to determine compression thresholds that preserve model outputs, while Thm. 3.1 highlights which layers are more tolerant to compression. The observed behaviour in Phi-2 is consistent with these predictions, supporting that the framework extends to more complex architectures.
> > >
> > > We will revise the manuscript to make these connections more explicit by including this analysis of Theorems 3.1 and 4.1 on the trained Phi-2 models to provide a link between the theoretical bounds and observed behaviour in a modern LLM setting.
> > >
> > > Lastly, we thank the reviewer again for their constructive feedback. We hope this clarifies the connection between theory and experiment, and demonstrates that the attack framework extends to architectures such as Phi-2, where Theorem 3.1 provides a local sensitivity analysis and Theorem 4.1 provides provable robustness bounds.

---

### Official Review · Reviewer_aW9B · 2026-03-12

**Soundness:** 3
**Presentation:** 3
**Significance:** 3
**Originality:** 3
**Overall Recommendation:** 4
**Confidence:** 3

**Summary:**

This work studies the weight perturbations in neural networks, and minimal weight perturbation needed to change predictions. The theoretical results are mainly two-fold. For single layer perturbations, the authors derive the closed-form equation of the minimal weight perturbation that alters the predicted label for certain samples. For multi-layer perturbations, they derive the lower bound of the perturbations that is required to alter the prediction, based on the Lipschitz smoothness assumption. The single-layer result indicates low-rank perturbation structures. In particular, the authors consider the low-rank approximation in the final layer. The class flips only when the input and output features lie in the subspace of residual weights. Motivated by this, the authors design a weight-modification-conditioned backdoor attack. SVD truncation would trigger such backdoor information, e.g., wrong classification.

**Compliance With Llm Reviewing Policy:**

Affirmed.

**Final Justification:**

The authors have addressed my concerns. I will keep the score.

**Key Questions For Authors:**

Please see above.

**Limitations:**

No. I think it would be better to discuss the assumptions such as the single-layer and Lipschitz assumptions.

**Strengths And Weaknesses:**

**Strengths**

1. The problem setting and the theoretical results are interesting. Low-rank approximations are popular now. It is of interest to study their potential effects on networks.

2. The bridge between the theoretical results and the practical compression backdoor attack is also interesting. This may raise practical safety concerns in related applications of compression.

**Weaknesses**

1. The low-rank theory and the empirical validation of the backdoor attack is limited to the single/final layer case. However, in most cases, users tend to compress the whole network. This degrades the practicality of the attack.

2. There seems to be some gap between the low-rank theory and the experiment in section 7. The attack in section 7 is obtained through a dedicated training objective rather than by directly instantiating the derived formula. Can we use full network compression as the compression map $g$ here?

---

> ### Author Rebuttal · Authors · 2026-03-31
>
> Thank you for your constructive feedback as well as supporting comments about the timeliness of low-rank approximation and bridging the theory and practical backdoor attack.  Speaking to the weaknesses and questions raised:
>
> * **Low-rank theory and experiments limited to single layer (Weakness 1).**
> Regarding the theory, yes the exact low-rank characterisation in Theorem 3.1 applies to the single-layer case.  While the formulae for the minima of a multi-layer perturbation can be stated in terms of the gradient being zero, standard KKT conditions, this results in a highly coupled set of equations that give limited insight can only be solved numerically.   In contrast, the more general margin-based robustness bound of Theorem 4.1 captures the effect of perturbations applied across multiple layers in a simple formula, but does so at the cost of less precision.  Figure 2 shows an example contrasting the single-layers vs multiple-layer perturbations which are surprisingly of a similar order.  One benefit of the single-layer result is that they can be used in a layer-wise manner to identify particularly sensitive layers, providing insight into where compression-induced vulnerabilities are most likely to arise. Though the specific low-rank theorem is final-layer only, Theorem 3.1 may be applied to any layer in this manner.
>
>     Additionally, in our initial experiments on smaller models, we found that applying low-rank compression across all layers was often overly aggressive, as even a small reduction in rank represents a large relative perturbation when layer dimensions are small. In contrast, methods such as pruning or quantization tend to be more conservative when applied globally.  To address this, we have extended our empirical evaluation in the rebuttal to include experiments on a larger LLM (Phi-2), where the hidden dimensions are significantly larger. In this setting, we observe that multi-layer low-rank compression becomes more meaningful, and we include results for both single-layer and multi-layer applications on this larger architecture. These results are also provided in **Table 1**in the response to review 3fhq.
>
>     We have clarified these distinctions and added the additional multi-layer low-rank LLM results in the revised manuscript.
>
>
> * **Backdoor training in Section 7 and connection to theory (Weakness 2).**
> We thank the reviewer for raising this point. The apparent gap reflects the different roles of the theoretical analysis and the experimental setup.
>
>     In Section 7, we first construct a model that exhibits compression-activated backdoor behaviour by training with a dedicated objective, effectively modelling an attacker. This step ensures that the desired behaviour is present in the trained network. We then consider the perspective of a defender, and apply our theoretical results to analyse such models, by characterising whether a given low-rank compression is sufficient to induce a change in the output.
>
>     In this sense, the theory is not intended to directly instantiate the attack, but rather to provide a principled framework for analysing and certifying when such behaviour can occur. In particular, models trained to exhibit this type of adversarial behaviour tend to be more sensitive to structured perturbations, and our bounds aim to capture the level of compression required to trigger such changes.
>
>     Regarding the use of full-network compression, this is indeed possible within our framework. While Theorem 3.1 provides a sharp characterisation for single-layer perturbations, Theorem 4.1 extends to perturbations applied across multiple or all layers via a Lipschitz-based bound. Theorem 3.1 may also be applied layer-wise to highlight layers which are particularly vulnerable to weight perturbations inducing changes in output. To demonstrate that an attack can be embedded when using low-rank on full network compression, we have conducted additional experiments on a larger LLM (Phi-2), where we apply low-rank compression across multiple layers and observe that the attack remains effective in this setting. We will clarify these distinctions and include the corresponding results in the revised manuscript.  The Phi-2 LLM was chosen as it was the largest general LLM network considered by Egashira et al. (https://arxiv.org/pdf/2405.18137) where they showed similar vulnerabilities to backdoor attacks through quantization.
>
> We thank the reviewers again for their constructive feedback and hope that our responses have addressed the concerns.

---

> > ### Author Rebuttal · Reviewer_aW9B · 2026-04-03
> >
> > Thanks for the authors' responses. I will keep my score.

---

### Official Review · Reviewer_XW9W · 2026-03-12

**Soundness:** 3
**Presentation:** 1
**Significance:** 2
**Originality:** 2
**Overall Recommendation:** 4
**Confidence:** 2

**Summary:**

This work is motivated by the sensitive adoption of network compression techniques. This work aims to quantify the relationship between weight perturbations and output changes in neural networks by deriving a closed-form expression under the assumption of locally invertible downstream maps. It establishes a general bound on the perturbation norm based on the Lipschitz-based analysis. This analysis can be extended to arbitrary architectures with non-linear activation functions. By applying these results, the authors showed empirically that low-rank compression can activate latent backdoors.

**Compliance With Llm Reviewing Policy:**

Affirmed.

**Final Justification:**

The authors second round of rebuttal fully solved my concern, so I decided to raise my score.

**Key Questions For Authors:**

1. There is a very abundant body of work on the sensitivity of neural networks and Lipschitz-based analysis of neural network robustness. Please clarify the novelty of this work compared to the existing literature, for example: https://arxiv.org/abs/1706.08498.

2. The authors claim that biased terms do not affect the overall results. However, would bias affect the local invertibility of the network?

3. When you validate Theorems 3.1 and 4.1, why did you choose LeakyReLU specifically in the synthetic numerical experiments? Does a change of activation functions affect the numerical results?

**Limitations:**

yes.

**Strengths And Weaknesses:**

Strength:
1. Theoretically, this paper gives an exact-form solution for the minimal weight perturbation, which shows novelty compared to previous work in the theoretical analysis of the sensitivity of neural networks.
2. The authors applied the theoretical results to the backdoor attacks and established a threshold for successful attacks.
3. The analysis connects the perturbation sensitivity to low-rank parameter changes.

Weakness:

1. There is no comparison with related work at all. There is a very abundant body of work on the sensitivity of neural networks and Lipschitz-based analysis of neural network robustness. Please clarify the novelty of this work compared to the existing literature, for example: https://arxiv.org/abs/1706.08498. Also, please compare the differences between this paper and other literature related to low-rank structure and perturbation analysis.

2. The analysis relies on a strong assumption that the layer-N is locally invertible. There is a need to clarify whether, under different activation functions/layers, this assumption is true.

---

> ### Author Rebuttal · Authors · 2026-03-31
>
> Thank you for your constructive feedback as well as supporting comments about the novelty of the theory and its application to perturbation sensitivity for backdoor attacks.  Speaking to the weaknesses and questions raised:
> * **Comparison with related work (Weakness 1 and Q1).** We thank the reviewer for this suggestion and agree that a clearer comparison with related work is needed. The revised manuscript strengthens this aspect by introducing a dedicated related work section and making connections to the existing literature more explicit.
>
>     The results nearest to our Theorem 3.1 are by Tsai et al. \url{https://arxiv.org/abs/2103.02200} who study generalisation and adversarial robustness under weight perturbations by deriving bounds on the pairwise class margin under norm-bounded perturbations to the network parameters.  Their Theorem 1 provides a bound on the output when a single layer is modified; this bound is  given in terms of the product of norms of the weights that follow the perturbed layer. This differs from our Theorem 3.1 by not providing the formulae for the perturbation and, more significantly, their bounds are far less precise as they are worst case over the change in each layer.  The bound in Tsai et al. Theorems 2 and 3 build from Theorem 1 and are again extremely pessimistic.
>
>     The results closest to our Theorem 4.1 are by Weng et al. \url{https://ojs.aaai.org/index.php/AAAI/article/view/6105/5961} which studies robustness to weight perturbations by computing certified regions in parameter space within which the network’s prediction is guaranteed to remain unchanged.  Conceptually, while Weng et al. characterise regions of guaranteed robustness, our Theorem 4.1 characterises the minimal perturbation required to break robustness expressed in closed form through a parameter-space Lipschitz constant. Our margin-based formulation is architecture-agnostic and provides a direct analogue of classical input-space robustness results, enabling a unified interpretation of parameter perturbations across layers.
>
>     There are numerous manuscripts that consider the sensitivity of the input-output map though Lipschitz and/or spectral-norm bounds.  Examples include: Bartlett et al., 2017, Tsuzuku et al., 2018; Fazlyab et al., 2019; and Pauli et al., 2021 and 2024.  These include margin-based and spectral-norm generalisation bounds as well as methods for certifying robustness or training stable networks via Lipschitz constraints, but none focus on the change in the parameter space which is our focus.
>
> * **Invertibility of the $N^{th}$-layer (Weakness 2).**
> We would like to clarify that the local invertibility assumption is not imposed on layer $N$ itself, but rather on the downstream map $h_{N:M}$. In particular, for the perturbed output $Y' = h_{N:M} (W_N' h_{1:N-1} (X))$, we assume that $h_{N:M}$ is locally invertible at $Z := W_N h_{1:N-1}(X)$, and that $\tilde{Y}$ lies within the corresponding local image. Under this assumption, both $h_{N:M}^{-1}(Y')$ and $h_{N:M}^{-1}(Y)$ are well-defined.
>
>     For ReLU networks, rank deficiency may arise when units are inactive (i.e. have negative pre-activations), since the corresponding derivatives are zero and certain directions are suppressed. The zero region of ReLU corresponds precisely to the source of non-invertibility: the mapping is locally constant along directions associated with inactive units, and therefore cannot be inverted. In addition, the inverse function theorem does not apply at activation boundaries where the network is not differentiable. Our analysis is intended to apply locally within regions where the activation pattern is fixed and the Jacobian is well-defined, which is standard in similar sensitivity analyses.
> * **Bias and invertibility for Theorem 3.1 (Qn 2).**
> For a single linear layer downstream map $h_{N:M}(z) = Wz + b = y$, the inverse is given by $h_{N:M}^{-1}(y) = W^\dagger (y - b)$, so invertibility depends only on the properties of $W$.  However, bias can indirectly influence invertibility in the presence of non-linear activations. For instance, with ReLU-type activations, the bias shifts the pre-activation and thus affects which neurons are active. Therefore, the local invertibility condition must still be verified under the induced activation pattern, as assumed in our analysis.
> * **Choice of LeakyReLU for numerical experiments (Qn 3).**
> The validation of Theorem 3.1 (Section 6.1) relies on the assumption that the downstream map after the perturbed layer is invertible. LeakyReLU was chosen due to it being invertible for suitable choices of $\alpha$ (in this case, we used the default value in the torch package $\alpha=0.01$) where $\mathrm{\texttt{LeakyReLU} }(x) := \max\{\alpha x, x\}$. We could have chosen a different activation, so long as invertibility of the downstream map was preserved.
>
> Lastly, we thank the reviewer for their feedback. We hope our clarifications address the concerns and clarify the contributions.

---

> > ### Author Rebuttal · Reviewer_XW9W · 2026-04-03
> >
> > Thank you to authors for their efforts in the detailed rebuttal. However, the many conditions needed for the invertibility of the downstream map makes me concerned about the scope of the proposed theory. I decided to maintain my score.

---

> > > ### Author Response · Authors · 2026-04-04
> > >
> > > We agree with the reviewer that the local invertibility assumption in Thm. 3.1 is restrictive and does not strictly hold for modern deep networks such as LLMs, where components such as ReLU, LayerNorm, and attention are not globally invertible. However, we find that the underlying framework of Thm. 3.1 remains practically useful when applied in a local, approximate sense to modern deep networks which we will briefly describe here, and we emphasise that Thm. 4.1 provides a weaker result not reliant on such assumptions.
> > >
> > > In practice, we do not apply Thm. 3.1 in its exact nonlinear form. Instead, we use a first-order local approximation, replacing the inverse of the downstream map with its Jacobian at the operating point. This yields a closed-form, gradient-based estimate of the minimum-norm perturbation, which we compute efficiently using autograd. Importantly, this formulation does not require invertibility of the downstream map; rather, it corresponds to a least-norm solution of the linearised system, which is equivalent to applying the Moore–Penrose pseudoinverse to the local Jacobian. This remains well-defined even when the Jacobian is rank-deficient.
> > >
> > > This approximation is valid in a local regime where perturbations are small and activation patterns remain stable, and may become less accurate under larger perturbations or when nonlinearities induce changes in the activation structure. We will clarify this distinction in the manuscript.
> > >
> > > Despite these limitations, we find that the resulting estimates are stable across samples when tested and align well with observed behaviour under compression. In particular, when applied to the LLM Phi-2 (Results given in Fig. 1: https://anonymous.4open.science/r/icml_results-3FD4/theorem1_min_perturbation_per_layer.pdf), the method identifies later layers (e.g. layer 30) as more sensitive to perturbations, while earlier layers exhibit greater robustness. These trends are consistent with the empirically observed compression behaviour, where later layers are more susceptible to inducing changes in the output. Therefore, Thm. 3.1 may be used to characterise safe compression levels for particular layers of trained deep networks on a validation set (i.e. Fig. 1 shows that Phi-2 may be safely compressed to rank 1900 on the validation set tested without causing a change in outputs).
> > >
> > > We emphasise that this use of Thm. 3.1 should be interpreted as a local sensitivity analysis tool, rather than a formal robustness certificate. In contrast, Thm. 4.1 does not rely on invertibility assumptions and provides a provable (albeit more conservative) robustness bound based on the Lipschitz constant, which applies more broadly to modern deep networks. The comparison to Thm. 4.1 in deep networks is also plotted on Fig. 1, and we see that this provides a more conservative robustness certificate where compression to rank 2100 is safe.
> > >
> > > We will revise the manuscript to clarify the role of this local approximation, and hope this demonstrates that, while Thm. 3.1 relies on restrictive assumptions in its exact form, its local formulation remains a useful and empirically validated tool for analysing layer-wise sensitivity in modern deep networks.

---

### Decision · Program_Chairs · 2026-04-30

**Decision:**

Accept (regular)

**Comment:**

The paper derives formulas for minimal weight perturbations, compares them with multilayer Lipschitz guarantees, and applies the analysis to low-rank activated backdoor attacks. It claims certifiable compression thresholds and empirical activation of latent backdoors under low-rank compression.

Reviewers found the theoretical problem timely and the exact-form analysis novel. They also highlighted the link to low-rank parameter changes and the bridge to compression-based backdoor attacks.

Main concerns were limited comparison with related work, strong assumptions such as local invertibility, and the single-layer or single-sample scope of the exact theory and experiments. One reviewer also found the experimental evidence too limited.

In rebuttal, the authors said they added clearer related-work discussion, clarified the single-layer scope, and revised Theorem 3.1 to include the feasibility condition. Post-rebuttal reactions remained mixed: one reviewer raised the score, one kept the score, and one said the evidence was still not strong enough.

Overall, the authors and reviewers engaged in substantive discussion, reaching broad consensus on both the contributions and limitations of the work as described above. The remaining open concern centers on the sufficiency of experimental validation, particularly for modern large-scale architectures; however, the authors made a concrete attempt to address this by conducting additional experiments on the Phi-2 (2.7B) LLM under multi-layer low-rank compression and pruning, which Reviewer 3fhq acknowledged as a useful step. Given that the theoretical framework offers a novel perspective on the relationship between low-rank compression and robustness—an increasingly important topic—and provides certifiable guarantees that complement largely empirical prior work in this area, I recommend acceptance.